# Neonatal birth trauma and associated factors in low and middle-income countries: A systematic review and meta-analysis

Beshada Zerfu Woldegeorgis[1]*, Amanuel Yosef Gebrekidan[2], Gizachew Ambaw Kassie[2], Gedion Asnake Azeze[3], Yordanos Sisay Asgedom[4], Henok Berhanu Alemu[1], Mohammed Suleiman Obsa[5]

1 School of Medicine, College of Health Sciences and Medicine, Wolaita Sodo University, Wolaita Sodo, Ethiopia, 2 School of Public Health, College of Health Sciences and Medicine, Wolaita Sodo University, Wolaita Sodo, Ethiopia, 3 Department of Midwifery, College of Medicine and Health Sciences, Hawassa University, Sidama Region, Ethiopia, 4 Department of Epidemiology, College of Health Sciences and Medicine, Wolaita Sodo University, Wolaita Sodo, Ethiopia, 5 Department of Anesthesia, Arsi University, Asella, Ethiopia

* beshadazerfu@gmail.com

**Data Availability Statement:** All relevant data are within the manuscript and its Supporting Information files

## Abstract

Neonatal birth trauma, although it has steadily decreased in industrialized nations, constitutes a significant health burden in low-resource settings. Keeping with this, we sought to determine the pooled cumulative incidence (incidence proportion) of birth trauma and identify potential contributing factors in low and middle-income countries. Besides, we aimed to describe the temporal trend, clinical pattern, and immediate adverse neonatal outcomes of birth trauma. We searched articles published in the English language in the Excerpta Medica database, PubMed, Web of Science, Google, African Journals Online, Google Scholar, Scopus, and in the reference list of retrieved articles. Literature search strategies were developed using medical subject headings and text words related to the outcomes of the study. The Joana Briggs Institute quality assessment tool was employed and articles with appraisal scores of seven or more were deemed suitable to be included in the meta-analysis. Data were analyzed using the random-effect Dersimonian-Laird model. The full search identified a total of 827 articles about neonatal birth trauma. Of these, 37 articles involving 365,547 participants met the inclusion criteria. The weighted pooled cumulative incidence of birth trauma was estimated at 34 per 1,000 live births (95% confidence interval (CI) 30.5 to 38.5) with the highest incidence observed in Africa at 52.9 per 1,000 live births (95% CI 46.5 to 59.4). Being born to a mother from rural areas (odds ratio (OR), 1.61; 95% CI 1.18 to 2.21); prolonged labor (OR, 5.45; 95% CI 2.30, 9.91); fetal malpresentation at delivery (OR, 4.70; 95% CI 1.75 to 12.26); shoulder dystocia (OR, 6.11; 95% CI 3.84 to 9.74); operative vaginal delivery (assisted vacuum or forceps extraction) (OR, 3.19; 95% CI 1.92 to 5.31); and macrosomia (OR, 5.06; 95% CI 2.76 to 9.29) were factors associated with neonatal birth trauma. In conclusion, we found a considerably high incidence proportion of neonatal birth trauma in low and middle-income countries. Therefore, early identification of risk factors and prompt decisions on the mode of delivery can potentially contribute to the decreased magnitude and impacts of neonatal birth trauma and promote the newborn's health.

**Funding:** The author(s) received no specific funding for this work

**Competing interests:** The authors have declared that no competing interests exist.

**Abbreviations:** CI, Confidence interval; OR, odds ratio; JBI, the Joana Briggs Institute; LMICs, Low and middle-income countries; MeSH, Medical subject headings; PRISMA, Preferred Reporting Items for Systematic Reviews and Meta-Analyses; PROSPERO, Prospective Register of Systematic Reviews; SDG, Sustainable Development Goals; US, United States.

## Introduction

Birth trauma, which is defined as a structural or functional impairment of a newborn's body due to mechanical forces sustained during labor, delivery, or neonatal resuscitation [1], has continued to be a significant source of jeopardy for the neonates and the family and also evokes concerns for healthcare providers [2].

According to the World Health Organization's eleventh revision of the International Classification of Diseases, the diagnosis of birth trauma includes extracranial traumatic injuries such as caput succedaneum, subgaleal hemorrhage, cephalohematoma, and facial or ocular or nasal injuries; intracranial hemorrhages such as intraventricular hemorrhage, subdural hemorrhage, subarachnoid hemorrhage, and epidural hemorrhage; peripheral nerves injuries such as brachial plexus injury, phrenic nerve injury, and facial nerve injury; injury to skeletons such as clavicular fracture, humeral fracture, rib fracture, and femurs fracture; soft tissue injuries such as lacerations, contusions, bruisings, and fat necrosis; and organ injuries such as liver, spleen, kidney, adrenals, and trachea [3].

Birth trauma impacts on the prognosis of neonates are protean, ranging from clinically trivial extracranial superficial lesions to significant causes of morbidity and mortality, such as intracranial hemorrhage [4]. Chang et al. [5] described that resulting hypotension from traumatic subgaleal hemorrhage can lead to significant morbidities such as severe auditory dysfunction, cerebral palsy, and renal vein thrombosis, and estimated neonatal mortality proportion ranging from 12% to 18% [6]. Furthermore, the Global Burden of Disease Study 2019 found that newborn encephalopathy caused by birth trauma was a major contributor to neonatal mortalities ranging from 5.4% to 96.4% in low-resource settings [7]. Although research into the economic impacts of birth trauma focused in low and middle-income countries (LMICs) is lacking, a study conducted in the United States (US) revealed that neonatal brachial plexus palsy hospital length of stay was 48% higher (3.6 days vs. 2.5 days) and hospitalization costs were twice ($4891 vs. $2241) as high as in neonates without the condition [8].

The United Nations Sustainable Development Goals (SDG) 3.2 envisaged ending preventable deaths of newborns for each country aiming to reduce neonatal mortality to less than 12 per 1000 live births [9]. Nevertheless, at the midway point towards 2030, the special edition progress report against the global SDG indicator framework indicated that the neonatal mortality was just 18 per 1000 live births on a global scale [10] and continued to pose a considerable challenge in the south Asian and Sub Saharan African countries [7]. Birth trauma and asphyxia remain preventable causes of neonatal death, which accounts for an estimated half of all under-five mortality [11]. As a result, neonatal health has become an important public health concern worldwide [12].

Data exist on the incidence of birth trauma are divergent according to geographic areas and level of neonatal and obstetrics healthcare and so far a plethora of studies have documented the true estimate of neonatal birth trauma in resource-rich settings [2]. For instance, a nationwide registry study of 1,203,434 neonates in Finland suggests the overall incidence of birth trauma decreased from 34 per 1,000 live births in 1997 to 16.6 per 1,000 live births in 2017 [13]. Similarly, in the United States, hospital birth data obtained from the National Inpatient Sample of the Healthcare Cost and Utilisation Project demonstrated an incidence of birth trauma of 24.3 per 1,000 live births in 2003 [14], and 31.1 per 1,000 live births in 2021 [15]. A Chinese study comprising 4,682 newborns in Peking University International Hospital

reported a birth trauma cumulative incidence of 42.9 per 1,000 live births [16]. In Canada, Muraca and colleagues determined that the incidence of birth trauma was 9.6 per 1,000 live births in a retrospective review of records of 1,326,191 live births from 2013 to 2019 [17].

In LMICs, estimates from individual studies suggest that the cumulative incidence of birth-related trauma ranges from an estimated 3 per 1,000 live births [18] to 869 per 1,000 live births [19] in Africa; 3 per 1,000 live births [20, 21] to 4.7 per 1,000 live births [22] in Asia and Pacific region; and 6.7 per 1,000 live births [23] to 87.3 per 1,000 live births [24] in the middle east. Operative vaginal deliveries such as vacuum extractions or forceps [25–34], fetal presentation other than vertex [25, 26, 30, 31, 35, 36], shoulder dystocia [28, 29, 31], gestational age [35], prolonged labor [27, 29, 31, 36], general anesthesia [35], and newborn birth weight greater than 4000 grams [26, 29, 30, 34, 36] have been found to increase the odds of sustaining birth trauma. Moreover, logistic regression analysis demonstrated that nulliparous women [25, 28], lack of or inadequate antenatal care follow-up [29, 30, 32, 36], gestational diabetes mellitus [36], residence in rural settings [25, 26, 30], maternal age [32, 35], and abnormal fetal heart rate patterns [25] have been correlated with a statistically significant increase in neonatal birth trauma.

To the best of our knowledge, there is no comprehensive, up-to-date, and accurate epidemiological data available regarding neonatal birth trauma in LMICs. Keeping with this, we aimed to seek answers to the following questions: (1) What is the pooled cumulative incidence of neonatal birth trauma in the LMICs? (2) What are the trends and clinical patterns of neonatal birth trauma in the LMICs? (3) What are epidemiologic risk factors correlated with neonatal birth trauma in the LMICs? and (4) What are the immediate/short-term adverse outcomes of neonatal birth trauma in the LMICs?

## Methods

### Protocol registration and reporting

We wrote the review protocol based on the Preferred reporting items for systematic review and meta-analysis protocols (PRISMA-P) 2015 [37]. Details of the protocol for this systematic review were registered on PROSPERO with registration number CRD42023445028. Moreover, the review methods were amended four times (last modified on 14/08/2023) after registration and these records were made public along with the revision notes. The study was reported following the Preferred Reporting Items for Systematic Reviews and Meta-Analyses (PRISMA) 2020 statement [38] (**S1 Checklist**).

### Eligibility criteria

To define inclusion criteria, we consulted the methodological guidance for systematic reviews of observational epidemiological studies reporting prevalence and cumulative incidence data [39]. Accordingly, the CoCoPop mnemonic (Condition, Context, and Population) was adhered to. Population/Participants: were neonates (infants from birth to 28 days); condition/domain: studies that clearly stated and defined the factors of interest based on the incidence, associated factors, and trend or outcomes of birth trauma were included; and context/settings: all observational epidemiological studies (cross-sectional, case-control, and cohort studies) restricted to low and middle-income countries were considered. Moreover, articles reported in the English language from inception to July 31, 2023, and published in international or domestic peer-reviewed journals were included. Studies without full-text access; articles that contained insufficient information on the outcomes of interest; studies not available as free full-text; findings from personal opinions; articles reported outside the scope of the outcome of interest; qualitative study design; case reports; case series; letters to editors; and unpublished data were excluded.

## Information sources and search strategy

To ensure complete coverage of the topic by accounting for variability between the indexing in each database, the search typically included the electronic bibliographic medical databases of Excerpta Medica database, PubMed, Web of Science, African Journals Online, Google Scholar, and Scopus. Furthermore, the reference lists of included studies were scanned to ensure literature saturation. For the advanced search, initially, we conducted a preliminary search in our title and identified relevant search terms from Google Scholar, Wikipedia, article, and Google for each concept, and then combined them in an advanced search using Boolean logic ("AND" and "OR"), double quotes and truncation. Moreover, filtrations were applied concerning language, research designs, and study settings. The search was double-blinded and conducted by two authors (BZW and AYG) from June 1, 2023, to July 31, 2023 (**Table 1**).

## Study selection

The articles that were found through comprehensive searches were exported to EndNote X7, where duplicate studies were then eliminated. Two authors (MSO and BZW) independently screened the titles and abstracts against eligibility criteria. The screened articles were then subjected to a full article review by two independent authors (YSA and GAA). Pre-specified criteria for inclusion in the review were followed to determine which records were relevant and should be included. Where additional information was required to answer queries regarding eligibility, other authors were involved as needed. Disagreements about whether a study should be included were resolved by discussion. Furthermore, the reasons for excluding the articles were recorded at each step

## Data extraction

Two authors (BZW and YSA) working independently excerpted the relevant data from studies by using a standardized Microsoft Excel spreadsheet. For data extraction, the Joana Briggs Institute (JBI) adopted data collection formats suitable for meta-analysis was used [40]. The data extraction format captured data on the following main components: information about data extraction from reports (name of data extractors, date of data extraction, and study identification number); study authors; year of publication of the article; study methods (study design, statistical analysis); participants and settings (regions and country from which study

**Table 1. PubMed's history and search details.**

| Search | Text Words (tw) | MeSH | | Query | Results |
|---|---|---|---|---|---|
| **#1** | incidence[tw], magnitude[tw], prevalence[tw], proportion[tw] | "Incidence" [MeSH Terms] | Incidence, Magnitude, Prevalence, Proportion | (incidence[MeSH Terms] OR magnitude [tw] OR incidence [tw] OR prevalence [tw] OR proportion [tw]) | 2,473,178 |
| **#2** | "birth trauma"[tw], "birth injur*"[tw] | "Birth injuries" [MeSH Terms] | Birth trauma, Birth injury, Birth injuries | ("birth injuries"[MeSH Terms] OR "birth trauma"[tw] OR "birth injur*"[tw]) | 7,410 |
| **#3** | newborns[tw], neonat*[tw], "live births"[tw] | "Infant, Newborn" [MeSH Terms] | Neonates, Newborns, Live births | ("infant, newborn"[MeSH Terms] OR newborns [tw] OR neonat*[tw] OR "live births"[tw]) | 848,237 |
| **#4** | "risk factors"[tw], "predisposing factors"[tw], "associated factors"[tw], determinants[tw], predictors[tw] | "Risk Factors" [MeSH Terms] | Associated factors, Risk factors, Predisposing factors, Determinants, Predictors | ("risk factors"[MeSH Terms] OR "risk Factors"[tw] OR "predisposing factors"[tw] OR "associated factors"[tw] OR determinants[tw] OR predictors [tw]) | 1,689,149 |
| **#1 AND #2 AND #3 AND #4** | | | | | 265 |

Abbreviations: MeSH, Medical subjects headings.

participants were recruited); information related to the pre-specified outcome domain in this systematic review (i.e birth trauma): measurement tool or instrument (including the definition of birth trauma); and information related to the results: for each study included in the quantitative analysis number of participants randomly assigned and included in the analysis, and the response rate. The reliability agreement among the data extractors was evaluated and verified using Cohan's kappa coefficient after data was recovered from 30% of the primary studies [41]. As a consequence, the kappa coefficient's strength of agreement was divided into five categories: low (0.20), fair (0.21–0.40), moderate (0.41–0.60), good (0.61–0.80), and virtually perfect agreement (0.81–1). A kappa statistic value of more than or equal to 0.5 was regarded as congruent and acceptable. In the case of disagreements between the two data extractors, a third author (MSO) was involved in adjudicating unresolved disagreements through discussion and re-checking of the original articles.

## Effect measures

The outcome of this systematic review and meta-analysis was neonatal birth trauma. As a result, the OR was the appropriate effect measure for meta-analysis of dichotomous outcome data [42]. The 'odds' refers to the ratio of the probability that the neonatal birth trauma occurred to the probability that it did not.

## Risk of bias (quality) assessment

The JBI critical appraisal checklist for studies reporting prevalence data [39], analytical cross-sectional and case-control study designs [43] were adhered to. Two authors (AYG and GAK) working independently carried out the quality assessment. Thus, studies conducted using a descriptive cross-sectional research design (n = 26) were evaluated against the following nine constructs: (Q1) Was the sample frame appropriate to address the target population? (Q2) Were study participants recruited appropriately? (Q3) Was the sample size adequate? (Q4) were the study subjects and setting described in detail? (Q5) Was data analysis conducted with sufficient coverage of the identified sample? (Q6) Were valid methods used for the identification of the condition? (Q7) Was the condition measured in a standard, reliable way for all participants? (Q8) Was there appropriate statistical analysis? and (Q9) Was the response rate adequate, and if not, was the low response rate managed appropriately? (**S1A Table**).

Articles that employed analytical cross-sectional study designs (n = 9) were critically appraised against the following eight questions: (Q1) Were the criteria for inclusion in the sample clearly defined? (Q2) Were the study subjects and the setting described in detail? (Q3) Was the exposure measured validly and reliably? (Q4) Were objective, standard criteria used for the measurement of the condition? (Q5) Were confounding factors identified? (Q6) Were strategies to deal with confounding factors stated? (Q7) Were the outcomes measured validly and reliably? (Q8) Was appropriate statistical analysis used? (**S1B Table**).

The remaining articles (n = 2), which employed the case-control study design, were evaluated against the following ten items: (Q1) Were the groups comparable other than the presence of disease in cases or the absence of disease in controls? (Q2) Were cases and controls matched appropriately? (Q3) Were the same criteria used for the identification of cases and controls? (Q4) Was exposure measured in a standard, valid, and reliable way? (Q5) Was exposure measured in the same way for cases and controls? (Q6) Were confounding factors identified? (Q7) Were strategies to deal with confounding factors stated? (Q8) Were outcomes assessed in a standard, valid, and reliable way for cases and controls? (Q9) Was the exposure period of interest long enough to be meaningful? (Q10) Was appropriate statistical analysis used? (**S1C Table**). For each question, the response option was, "yes", "no" or "unclear". The total score

was determined by counting the "yes" responses to each question and adding them. In all scenarios, articles with appraisal scores of seven or more were deemed suitable to be included in the meta-analysis. When disagreements arose, they were settled by consulting with a third independent author (HBA)

### Data synthesis methods

Extracted data were imported from Microsoft Excel 2010 into Stata 16 MP version for analysis. The presence and degree of variability (inconsistency/heterogeneity) among individual studies were evaluated graphically (present when the uncertainty interval for the results of individual studies generally depicted in forest plots using the horizontal lines have poor overlap) and more formally, using statistical methods (the $X^2$ test, included in the forest plots, significant level:0.1; Higgins and Thompson's $I^2$ statistics: 0–25%: low heterogeneity; 25–50%: moderate heterogeneity; 50–75%: high heterogeneity; 75–100%: very high heterogeneity) [42]. We employed the random-effect meta-analysis model to estimate Der Simonian and Laird's pooled effect as considerable statistical heterogeneity was observed (Higgins and Thompson's $I^2$ statistics was $\geq$ 50% and $P$.value was $\leq$ 0.1) in the fixed-effect meta-analysis model. Moreover, subgroup analyses (based on year of publication, and study region), meta-regression (based on year of publication, and sample size), and sensitivity analyses were performed.

To evaluate the presence of small study effects, publication bias was explored through statistical methods (Egger test: significant at $P \leq 0.05$) and graphical approaches (funnel plots) [44]. The symmetrical distribution of the points about the summary effect size is an indication of the absence of a possible small-study effect or publication bias. However, any asymmetrical distribution of the points (the typical pattern in the presence of small-study effects is a prominent asymmetry at the bottom that progressively disappears as we move up to larger studies) may support the presence of a possible small-study effect or publication bias [44, 45]. Due to the presence of publication biases for the pooled cumulative incidence of neonatal birth trauma, we conducted the non-parametric trim-and-fill method of Duval and Tweedie. Regarding the determinant factors variables with $P \leq 0.05$ were considered statistically significant and the strength of the association was presented by OR with a corresponding 95% confidence interval (CI). Tables, graphs and text narration were used to present results.

## Results

### Search results

The search identified 827 articles. Due to duplication, 540 articles were removed. The remaining 287 were screened based on their title and abstract, with 235 being removed as unrelated to our domain. Fifty-two full-text articles were evaluated against eligibility criteria and 26 of them were removed (due to different outcomes,n = 2; different settings, n = 18, and reported only single birth injury type, n = 6). In addition, citation searching identified 11 articles. Finally, 37 articles were included in the quantitative synthesis (**Fig 1**).

### Study characteristics

There were 365,547 live births from Ethiopia [25, 26, 29, 30, 32, 36], Ghana [18, 46], Nigeria [19, 47–54], Cameroon [55], Niger [56], India [20–22, 57, 58], Israel [59], Iran [24, 28, 33, 60–64], Pakistan [65], Thailand [27], Iraq [31] and Saudi Arabia [23, 66]. Nineteen of 37 studies (51.4%) were conducted in African countries [18, 19, 25, 26, 29, 30, 32, 36, 46–56]. The remaining 18 studies (48.6%) were conducted in Asian countries [20–24, 27, 28, 31, 33, 57–66]. The authors employed cross-sectional [18–30, 32, 36, 46–66] and case-control [31, 33]

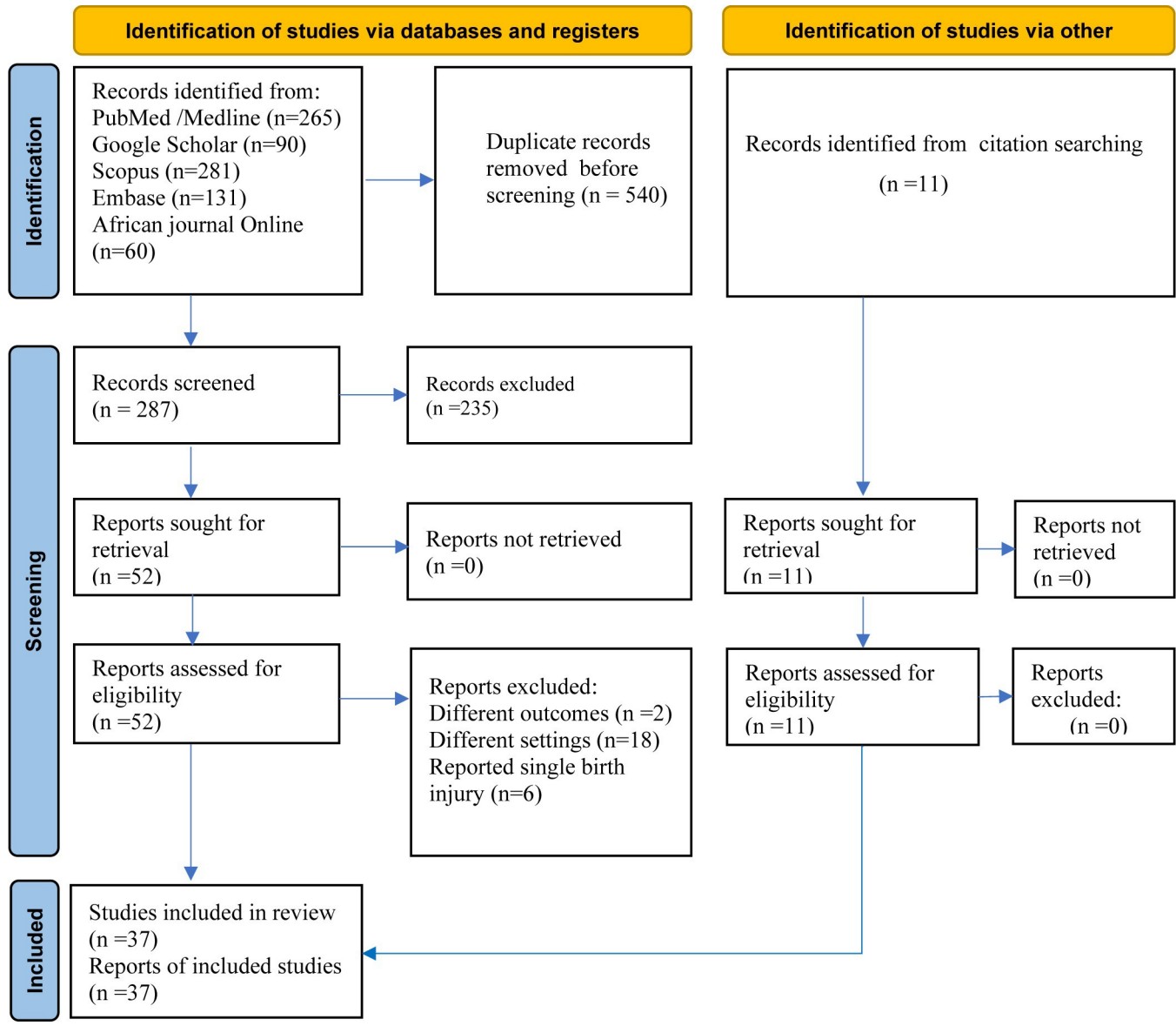

**Fig 1. PRISMA flow diagram for identification and selection of articles.**

observational epidemiological study designs. The study sizes ranged from 131 [33] to 118,280 [59] live births. A standardized data abstraction tool [18–30, 32, 33, 46–66] and interviewer-administered structured questionnaires with a data abstraction tool [29, 31, 36, 49–51] were employed for data collection. The year of publication ranges from 1985 [61] to 2023 [30, 33, 36]. A total of 6,429 newborns incurred at least one traumatic birth injury. A wide range of estimates with the lowest cumulative incidence in Ghana at 2.99 per 1,000 live births [18] and the highest in Ethiopia at 168.7 per 1,000 live births [30] were reported (**Table 2**).

## Incidence of neonatal birth trauma

In this epidemiological review of studies suited for meta-analysis, incidence data were abstracted from 35 articles [18–30, 32, 36, 46–61, 63–66] involving 365,547 live births. The

**Table 2. Summary of studies included in the systematic review and meta-analysis.**

| SN | Authors /reference | Year | Continent | Country | Study design | Data collection tool | Study size | Neonatal birth trauma | | Quality |
|----|--------------------|------|-----------|---------|--------------|----------------------|------------|-------|------------------------------|---------|
| | | | | | | | | Count | Incidence per 1,000 live births | |
| 1 | Tesfaye et al [25] | 2016 | Africa | Ethiopia | Analytical CS | Extraction | 272 | 42 | 154.4 | 8 |
| 2 | Yemane et al [26] | 2019 | Africa | Ethiopia | Analytical CS | Extraction | 717 | 88 | 122.7 | 8 |
| 3 | Biset et al [29] | 2022 | Africa | Ethiopia | Analytical CS | Interview & Extraction | 594 | 78 | 131.3 | 8 |
| 4 | Tolosa et al [30] | 2023 | Africa | Ethiopia | Analytical CS | Extraction | 492 | 83 | 168.7 | 8 |
| 5 | Belay et al [32] | 2022 | Africa | Ethiopia | Analytical CS | Extraction | 1,315 | 220 | 167.3 | 8 |
| 6 | Mah et al [55] | 2017 | Africa | Cameron | Descriptive CS | Extraction | 14,284 | 263 | 18.4 | 8 |
| 7 | Abdul-mumin et al [46] | 2021 | Africa | Ghana | Descriptive CS | Extraction | 5,590 | 205 | 36.7 | 8 |
| 8 | Pius et al [49] | 2018 | Africa | Nigeria | Descriptive CS | Interview & Extraction | 1,071 | 61 | 57.0 | 8 |
| 9 | West and Okari [50] | 2021 | Africa | Nigeria | Descriptive CS | Interview & Extraction | 5,692 | 39 | 6.9 | 8 |
| 10 | Emeka et al [51] | 2019 | Africa | Nigeria | Descriptive CS | Interview & Extraction | 1,735 | 19 | 11.0 | 8 |
| 11 | Warke et al [20] | 2012 | Asia | India | Descriptive CS | Extraction | 5,837 | 19 | 3.3 | 9 |
| 12 | Linder et al [59] | 2012 | Asia | Israel | Analytical CS | Extraction | 118,280 | 2,876 | 24.3 | 8 |
| 13 | Zama et al [22] | 2020 | Asia | India | Descriptive CS | Extraction | 850 | 100 | 117.7 | 8 |
| 14 | Phuengphaeng et al [27] | 2022 | Asia | Thailand | Analytical CS | Extraction | 15,209 | 81 | 5.3 | 8 |
| 15 | Borna et al [63] | 2009 | Asia | Iran | Analytical CS | Extraction | 3596 | 148 | 41.2 | 8 |
| 16 | Mosavat and Zamani [64] | 2008 | Asia | Iran | Descriptive CS | Extraction | 3340 | 27 | 8.1 | 9 |
| 17 | Abedzadeh-Kalahroudi et al [28] | 2015 | Asia | Iran | Analytical CS | Extraction | 7154 | 161 | 22.5 | 7 |
| 18 | Shanthi et al [21] | 2022 | Asia | India | Descriptive CS | Extraction | 12,221 | 40 | 3.3 | 8 |
| 19 | Awari et al [23] | 2003 | Asia | Saudi Arabia | Descriptive CS | Extraction | 31,028 | 208 | 6.7 | 8 |
| 20 | Rezaie et al [24] | 2009 | Asia | Iran | Descriptive CS | Extraction | 2005 | 175 | 87.3 | 8 |
| 21 | Enyindah et al [56] | 2005 | Africa | Niger | Descriptive CS | Extraction | 16631 | 50 | 3.0 | 8 |
| 22 | Adegbehingbe et al [19] | 2007 | Africa | Nigeria | Descriptive CS | Extraction | 137 | 119 | 86.9 | 8 |
| 23 | Osinaike et al [53] | 2017 | Africa | Nigeria | Descriptive CS | Extraction | 134 | 90 | 671.6 | 8 |
| 24 | Fabamwo et al [48] | 2006 | Africa | Nigeria | Descriptive CS | Extraction | 7200 | 44 | 6.1 | 8 |
| 25 | Njokanma and Kehinde [52] | 2002 | Africa | Nigeria | Descriptive CS | Extraction | 2941 | 50 | 17.0 | 8 |
| 26 | Okoro and Oriji [54] | 2018 | Africa | Nigeria | Descriptive CS | Extraction | 14814 | 107 | 7.2 | 8 |
| 27 | Danso and Shaka [18] | 1999 | Africa | Ghana | Descriptive CS | Extraction | 46113 | 138 | 3.0 | 8 |
| 28 | Uchenna et al [47] | 2021 | Africa | Nigeria | Descriptive CS | Extraction | 1920 | 46 | 24.0 | 8 |
| 29 | Gorashi et al [60] | 2005 | Asia | Iran | Descriptive CS | Extraction | 7660 | 102 | 14.1 | 8 |

*(Continued)*

**Table 2.** (Continued)

| SN | Authors /reference | Year | Continent | Country | Study design | Data collection tool | Study size | Neonatal birth trauma | | Quality |
|----|--------------------|------|-----------|---------|--------------|----------------------|------------|-------|-----------------------------------|---------|
| | | | | | | | | Count | Incidence per 1,000 live births | |
| 30 | Esmailpour et al [61] | 1985 | Asia | Iran | Descriptive CS | Extraction | 13117 | 141 | 10.8 | 8 |
| 31 | Ray et al [58] | 2006 | Asia | India | Descriptive CS | Extraction | 4741 | 73 | 15.4 | 8 |
| 32 | Prabhu et *al* [57] | 2017 | Asia | India | Descriptive CS | Extraction | 12735 | 283 | 22.2 | 8 |
| 33 | Benjamin et *al* [66] | 1993 | Asia | Saudi Arabia | Descriptive CS | Extraction | 2222 | 57 | 25.7 | 8 |
| 34 | Shabbir et *al [65]* | 2014 | Asia | Pakistan | Descriptive CS | Extraction | 3596 | 148 | 41.2 | 8 |
| 35 | Hameed and Izzet [31] | 2010 | Asia | Iraq | Case control | Extraction | 200 | NA | NA | 9 |
| 36 | Basiri et al [33] | 2023 | Asia ' | Iran | Case control | Extraction | 131 | NA | NA | 9 |
| 37 | Tibebu et al [36] | 2023 | Africa | Ethiopia | Analytical CS | Interview & Extraction | 373 | 48 | 128.7 | 8 |

Abbreviations: CS, cross-sectional; NA, not applicable

weighted pooled cumulative incidence of neonatal birth trauma was estimated to be 34 per 1,000 live births (95% CI 30.5 to 38.5) (**Fig 2**).

**Heterogeneity.** Due to the observed very high statistical heterogeneity (i.e. Hiddigns and Thomsons $I^2$ statistics = 99.3%, and $P < 0.001$), we conducted meta-regression and subgroup meta-analysis to explore the sources of statistical heterogeneity. As described in **Table 3**, the study size and publication year were not found to be the cause of the statistical heterogeneity.

Subgroup meta-analysis based on the region demonstrated that the cumulative incidence of neonatal birth trauma in Africa, 52.9 per 1,000 live births (95% CI 46.5 to 59.4), was more than two-fold higher than in the Middle East, 24.9 per 1,000 live births (95% CI 17.6 to 32.3), and about three-fold higher than in Asia and the Pacific countries, 18 per 1,000 live births (95% CI 12 to 23.9) (**Fig 3**).

According to the year of publication, the cumulative incidence of neonatal birth trauma was found to be 45.5 per 1,000 live births (95% CI 39.1 to 51.8) in 2015 and beyond (**Fig 4**).

**Sensitivity meta-analysis.** A leave-out-one sensitivity analysis was conducted to assess the impact of each study on the pooled incidence of neonatal birth trauma while gradually excluding each study. Results showed that the combined effects did not significantly change as a result of the excluded study (**Table 4**).

**Publication bias.** To explore whether there is a possibility of small-study effects, we examined the distribution of studies about the summary effect sizes using a graph. In this case, the funnel plot demonstrated a prominent asymmetrical distribution (**Fig 5**).

Moreover, the Egger linear regression test was statistically significant (t = 3.11; $P = 0.004$) further corroborating the presence of evidence of small study effects. The counter-enhanced funnel plot (**Fig 6A**) showed that small studies were found in non-statistical significance (white area). Thus, the asymmetry might have been caused by publication bias. The metric inverse counter-enhanced funnel plot (**Fig 6B**) also revealed the same.

When evaluated against the Egger© regression test, the estimated bias coefficient was 10.53967 with a standard error of 1.953324, a *P* value of < 0.001, and 95% CI 6.6 to 14.5. The test thus provides strong evidence for the presence of a small study effect. In addition, as

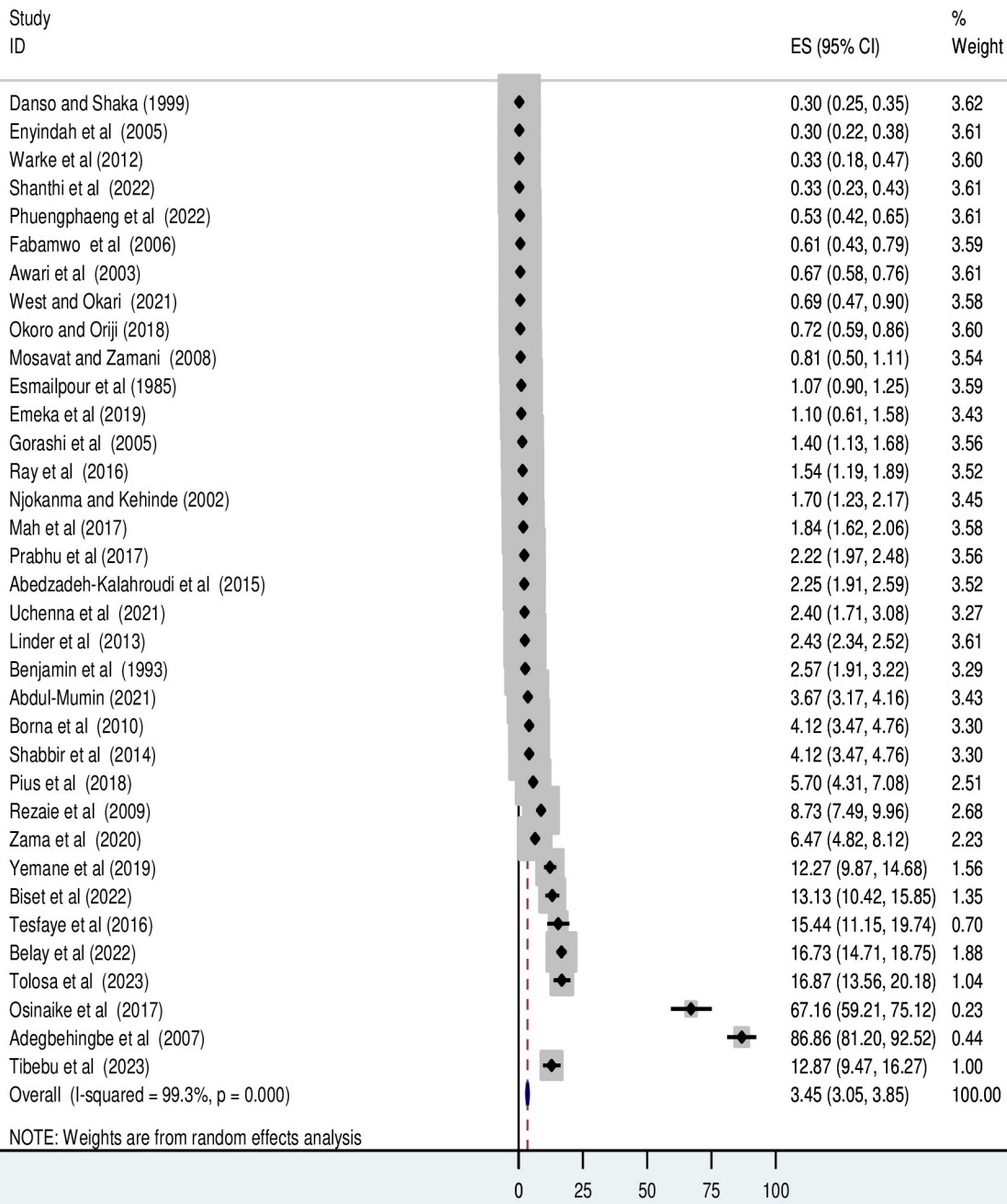

**Fig 2. Forest plot for the pooled cumulative incidence of neonatal birth trauma.**

**Table 3. Meta-regression analysis of factors affecting between-study heterogeneity.**

| Covariates | Regression coefficient | Standard error | t | P>|t| | 95% confidence interval | |
|---|---|---|---|---|---|---|
| Study size | -.0001549 | .0001482 | -1.05 | 0.304 | -.0004568 | .000147 |
| Year of publication | .1047447 | .3460033 | 0.30 | 0.764 | -.6000409 | .8095304 |

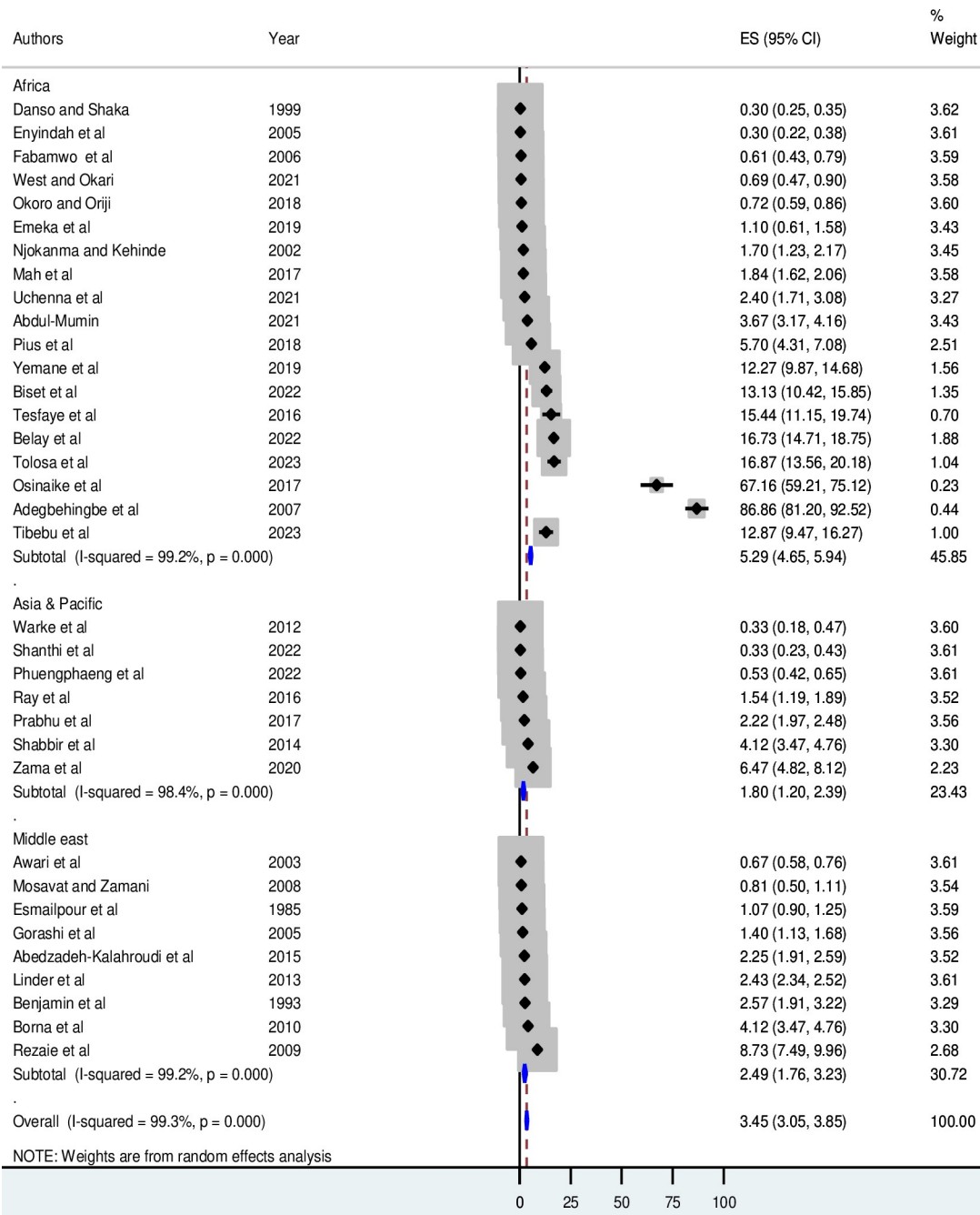

**Fig 3. Forest plot for subgroup analyses by region.**

shown in Fig 7 while only two estimates just touched the regression line, the majority of the data points were below the regression line.

We also carried out the non-parametric trim-and-fill method of Duval and Tweedie, tests for funnel-plot asymmetry, which provides a way to assess the impact of missing studies because of publication bias on the meta-analysis. Thus, the meta-trim analysis demonstrated the presence of 15 unpublished studies (**Fig 8**).

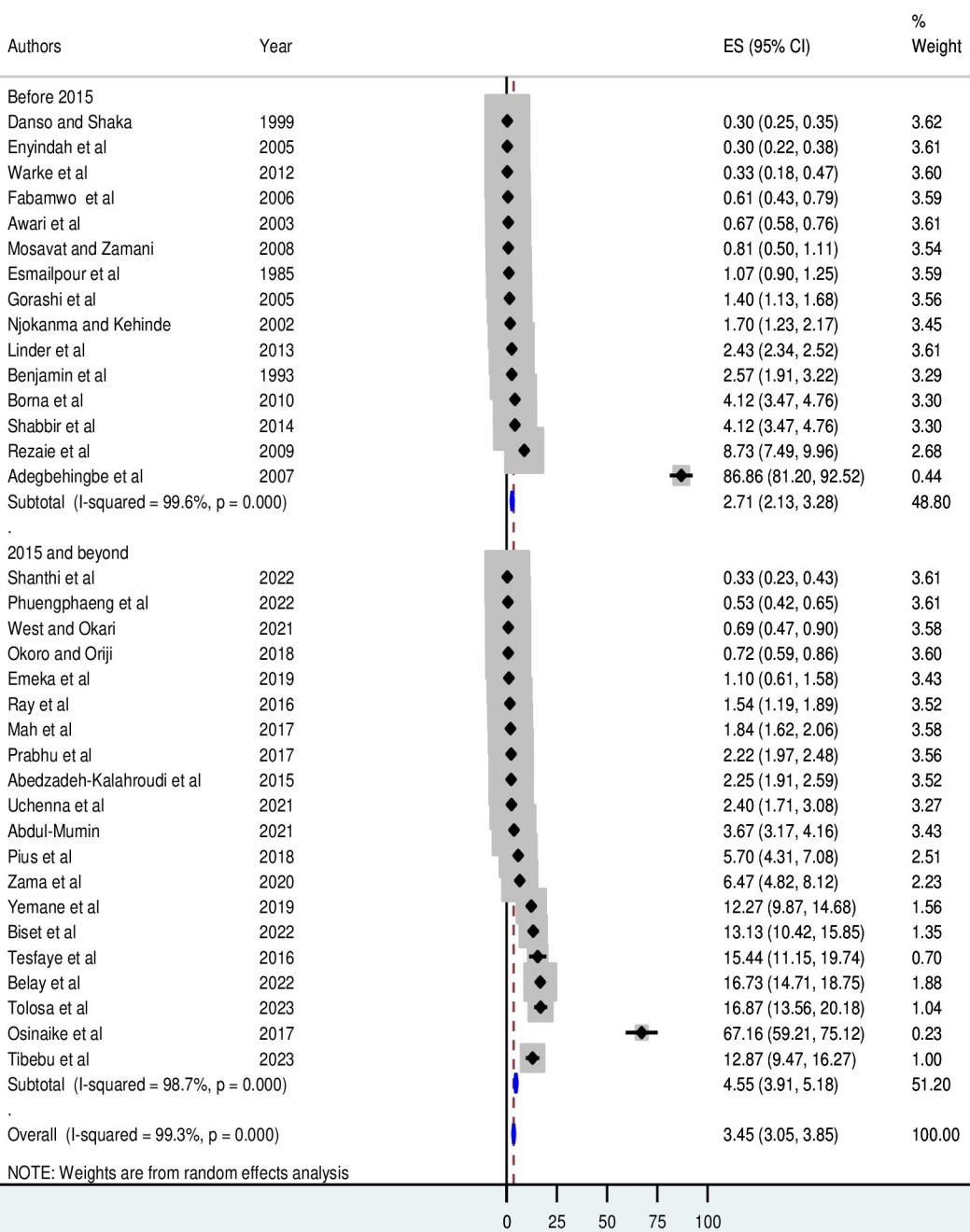

| Authors | Year | ES (95% CI) | % Weight |
|---|---|---|---|
| **Before 2015** | | | |
| Danso and Shaka | 1999 | 0.30 (0.25, 0.35) | 3.62 |
| Enyindah et al | 2005 | 0.30 (0.22, 0.38) | 3.61 |
| Warke et al | 2012 | 0.33 (0.18, 0.47) | 3.60 |
| Fabamwo et al | 2006 | 0.61 (0.43, 0.79) | 3.59 |
| Awari et al | 2003 | 0.67 (0.58, 0.76) | 3.61 |
| Mosavat and Zamani | 2008 | 0.81 (0.50, 1.11) | 3.54 |
| Esmailpour et al | 1985 | 1.07 (0.90, 1.25) | 3.59 |
| Gorashi et al | 2005 | 1.40 (1.13, 1.68) | 3.56 |
| Njokanma and Kehinde | 2002 | 1.70 (1.23, 2.17) | 3.45 |
| Linder et al | 2013 | 2.43 (2.34, 2.52) | 3.61 |
| Benjamin et al | 1993 | 2.57 (1.91, 3.22) | 3.29 |
| Borna et al | 2010 | 4.12 (3.47, 4.76) | 3.30 |
| Shabbir et al | 2014 | 4.12 (3.47, 4.76) | 3.30 |
| Rezaie et al | 2009 | 8.73 (7.49, 9.96) | 2.68 |
| Adegbehingbe et al | 2007 | 86.86 (81.20, 92.52) | 0.44 |
| Subtotal (I-squared = 99.6%, p = 0.000) | | 2.71 (2.13, 3.28) | 48.80 |
| . | | | |
| **2015 and beyond** | | | |
| Shanthi et al | 2022 | 0.33 (0.23, 0.43) | 3.61 |
| Phuengphaeng et al | 2022 | 0.53 (0.42, 0.65) | 3.61 |
| West and Okari | 2021 | 0.69 (0.47, 0.90) | 3.58 |
| Okoro and Oriji | 2018 | 0.72 (0.59, 0.86) | 3.60 |
| Emeka et al | 2019 | 1.10 (0.61, 1.58) | 3.43 |
| Ray et al | 2016 | 1.54 (1.19, 1.89) | 3.52 |
| Mah et al | 2017 | 1.84 (1.62, 2.06) | 3.58 |
| Prabhu et al | 2017 | 2.22 (1.97, 2.48) | 3.56 |
| Abedzadeh-Kalahroudi et al | 2015 | 2.25 (1.91, 2.59) | 3.52 |
| Uchenna et al | 2021 | 2.40 (1.71, 3.08) | 3.27 |
| Abdul-Mumin | 2021 | 3.67 (3.17, 4.16) | 3.43 |
| Pius et al | 2018 | 5.70 (4.31, 7.08) | 2.51 |
| Zama et al | 2020 | 6.47 (4.82, 8.12) | 2.23 |
| Yemane et al | 2019 | 12.27 (9.87, 14.68) | 1.56 |
| Biset et al | 2022 | 13.13 (10.42, 15.85) | 1.35 |
| Tesfaye et al | 2016 | 15.44 (11.15, 19.74) | 0.70 |
| Belay et al | 2022 | 16.73 (14.71, 18.75) | 1.88 |
| Tolosa et al | 2023 | 16.87 (13.56, 20.18) | 1.04 |
| Osinaike et al | 2017 | 67.16 (59.21, 75.12) | 0.23 |
| Tibebu et al | 2023 | 12.87 (9.47, 16.27) | 1.00 |
| Subtotal (I-squared = 98.7%, p = 0.000) | | 4.55 (3.91, 5.18) | 51.20 |
| . | | | |
| Overall (I-squared = 99.3%, p = 0.000) | | 3.45 (3.05, 3.85) | 100.00 |
| NOTE: Weights are from random effects analysis | | | |

0    25    50    75    100

**Fig 4. Forest plot for subgroup analyses by year of publication.**

## Trends and patterns of neonatal birth trauma

Fig 9 describes the trend of neonatal birth trauma. The highest cumulative incidence of neonatal birth trauma was documented in the year 2023 (168.7 cases per 1,000 live births).

Regarding the patterns, a wide gamut of birth trauma ranging from minor and self-limiting to severe trauma that had contributed to significant neonatal morbidity and mortality were noted. The most incident cases of birth trauma were cephalohematoma (an estimated 404 per 1,000 live births) followed by clavicular bone fracture (207 per 1,000 live births), and subgaleal

**Table 4. Sensitivity analysis of pooled cumulative incidence with each study removed one by one.**

| SN | Study omitted | Year of publication | Estimate per 1,000 live births | 95% confidence interval | |
|---|---|---|---|---|---|
| 1 | Tesfaye et al [25] | 2016 | 33.6 | 29.6 | 37.6 |
| 2 | Yemane et al [26] | 2019 | 32.9 | 29.0 | 36.9 |
| 3 | Biset et al [29] | 2022 | 33.0 | 29.1 | 37.0 |
| 4 | Tolosa et al [30] | 2023 | 32.9 | 29.0 | 36.9 |
| 5 | Belay et al [32] | 2022 | 31.5 | 27.6 | 35.4 |
| 6 | Mah et al [55] | 2017 | 35.2 | 31.1 | 39.2 |
| 7 | Abdul-mumin et al [46] | 2021 | 34.2 | 30.2 | 38.2 |
| 8 | Pius et al [49] | 2018 | 33.9 | 29.8 | 37.9 |
| 9 | West and Okari [50] | 2012 | 35.8 | 31.7 | 39.9 |
| 10 | Emeka et al [51] | 2019 | 35.4 | 31.4 | 39.5 |
| 11 | Warke et al [20] | 2012 | 36.2 | 32.0 | 40.3 |
| 12 | Linder et al [59] | 2012 | 32.3 | 28.6 | 35.9 |
| 13 | Zama et al [22] | 2020 | 33.7 | 29.7 | 37.8 |
| 14 | Phuengphaeng et al [27] | 2022 | 36.4 | 32.2 | 40.6 |
| 15 | Borna et al [63] | 2009 | 34.1 | 30.1 | 38.1 |
| 16 | Mosavat and Zamani [64] | 2008 | 35.6 | 31.5 | 39.7 |
| 17 | Abedzadeh-Kalahroudi [28] | 2015 | 34.9 | 30.9 | 40.0 |
| 18 | Shanthi et al [21] | 2022 | 36.6 | 32.4 | 40.9 |
| 19 | Awari et al [23] | 2003 | 36.9 | 32.7 | 41.2 |
| 20 | Rezaie et al [24] | 2009 | 32.7 | 28.7 | 36.7 |
| 21 | Enyindah et al [56] | 2005 | 37.0 | 32.7 | 41.3 |
| 22 | Adegbehingbe et al [19] | 2007 | 29.3 | 25.7 | 33.0 |
| 23 | Osinaike et al [53] | 2017 | 32.5 | 28.6 | 36.4 |
| 24 | Fabamwo et al [48] | 2006 | 35.9 | 31.8 | 40.1 |
| 25 | Njokanma and Kehinde [52] | 2002 | 35.2 | 31.1 | 39.2 |
| 26 | Okoro and Oriji [54] | 2018 | 36.2 | 32.0 | 40.3 |
| 27 | Danso and Shaka [18] | 1999 | 38.1 | 33.6 | 42.6 |
| 28 | Uchenna et al [47] | 2021 | 34.9 | 30.8 | 38.9 |
| 29 | Gorashi et al [60] | 2005 | 35.9 | 31. 3 | 39.5 |
| 30 | Esmailpour et al [61] | 1985 | 35.8 | 31.6 | 39.9 |
| 31 | Keshtkaran et al [62] | 2007 | 34.4 | 30.4 | 38.4 |
| 32 | Ray et al [58] | 2006 | 35.3 | 31.2 | 39.3 |
| 33 | Prabhu et al [57] | 2017 | 34.9 | 30.8 | 38.9 |
| 34 | Benjamin et al [66] | 1993 | 34.8 | 30.7 | 38.8 |
| 35 | Shabbir et al [65] | 2014 | 34.1 | 30.1 | 38.1 |
| 36 | Tibebu et al [36] | 2023 | 33.5 | 29.5 | 37.5 |
| Combined | | | 34.5 | 30.5 | 38.5 |

hemorrhage (99 per 1,000 live births). Amongst the peripheral nerve injuries, brachial plexus injury (Erb's Duchenne or Klumpke's palsy) was estimated at 98.6 per 1,000 live births (**Fig 10**).

## Adverse outcomes of neonatal birth trauma

In the studies that assessed immediate adverse outcomes [26, 46, 49–52, 59], a neonatal death incidence of 5.3% [51] to 28% [52] was reported. Profound neonatal hypovolemia secondary to the most clinically significant and potentially life-threatening injury,subgaleal hemorrhage [26, 46, 50, 52], and sepsis [51] were important complications of birth trauma that contributed

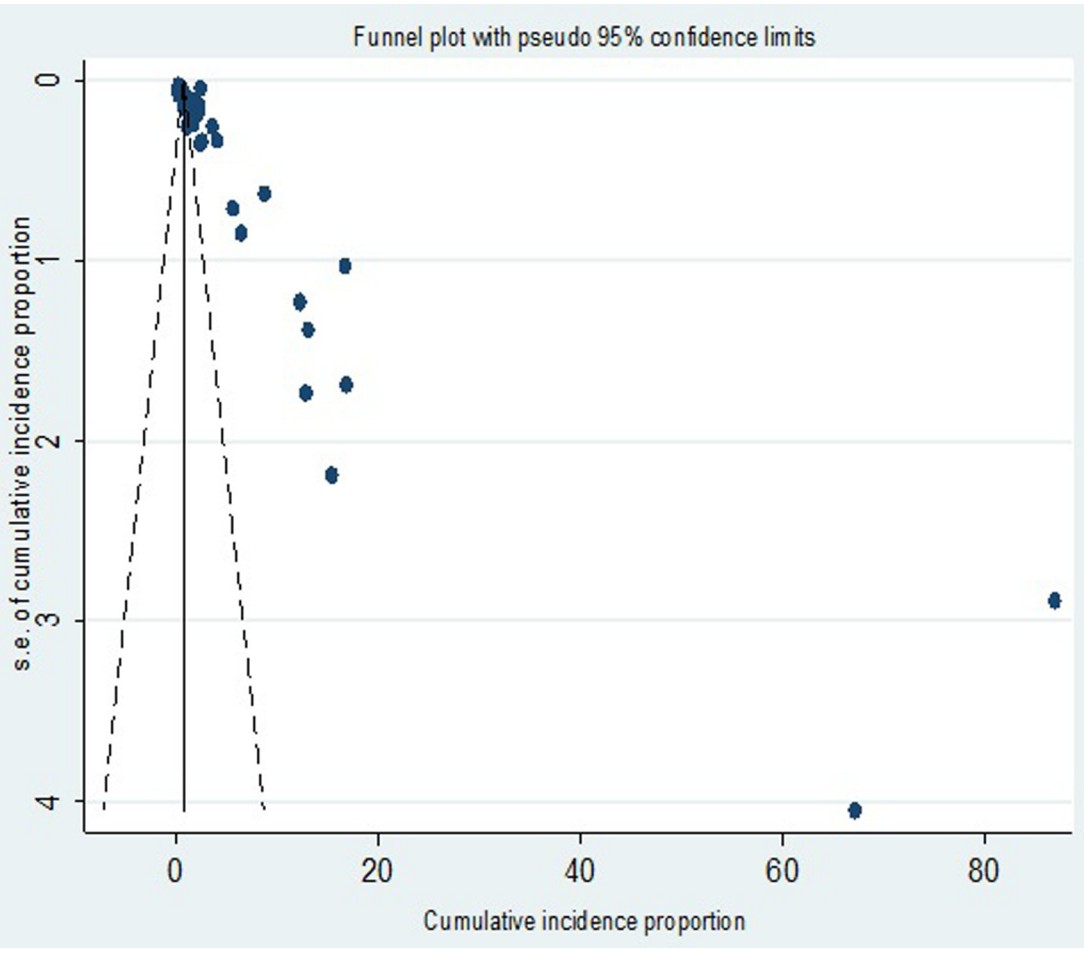

**Fig 5. Funnel plot for publication bias.**

to neonatal death. Another reported adverse outcome included anemia and hyperbilirubine-mia that required blood transfusion, and phototherapy respectively [26, 49, 59] (**Table 5**).

### Factors associated with neonatal birth trauma

Eleven of the 37 studies (29.7%) evaluated the associations of various exposure variables with neonatal birth trauma. Accordingly, place of residence; labor, fetal presentation, shoulder dystocia, birth weight, and mode of delivery were significantly and positively associated with increased odds of birth trauma. To begin with, neonates born to mothers living in rural areas had 61% higher odds [OR,1.61; 95% CI, 1.18 to 2.21; Higgins and Thompson's $I^2$ statistics = 0.0%; Egger's test for small-study effects = 0.110] of experiencing birth trauma compared to their counterparts. Prolonged labor was associated with 5.45-fold increased odds of neonatal birth trauma [OR, 5.45; 95% CI,2.30, 9.91; Higgins and Thompson's $I^2$ statistics = 71.8%; Egger's test for small-study effects = 0.498].

Abnormal fetal presentation at the time of vaginal delivery [OR, 4.70; 95% CI, 1.75 to 12.26; Higgins and Thompson's $I^2$ statistics = 89.2%; Egger's test for small-study effects = 0.885], and shoulder dystocia [OR, 6.11; 95% CI, 3.84 to 9.74; Higgins and Thompson's $I^2$ statistics = 0.0%; Egger's test for small-study effects = 0.358] were associated with 4.70 and 6.11times higher odds of sustaining neonatal birth trauma. Compared to cesarean delivery, assisted vaginal

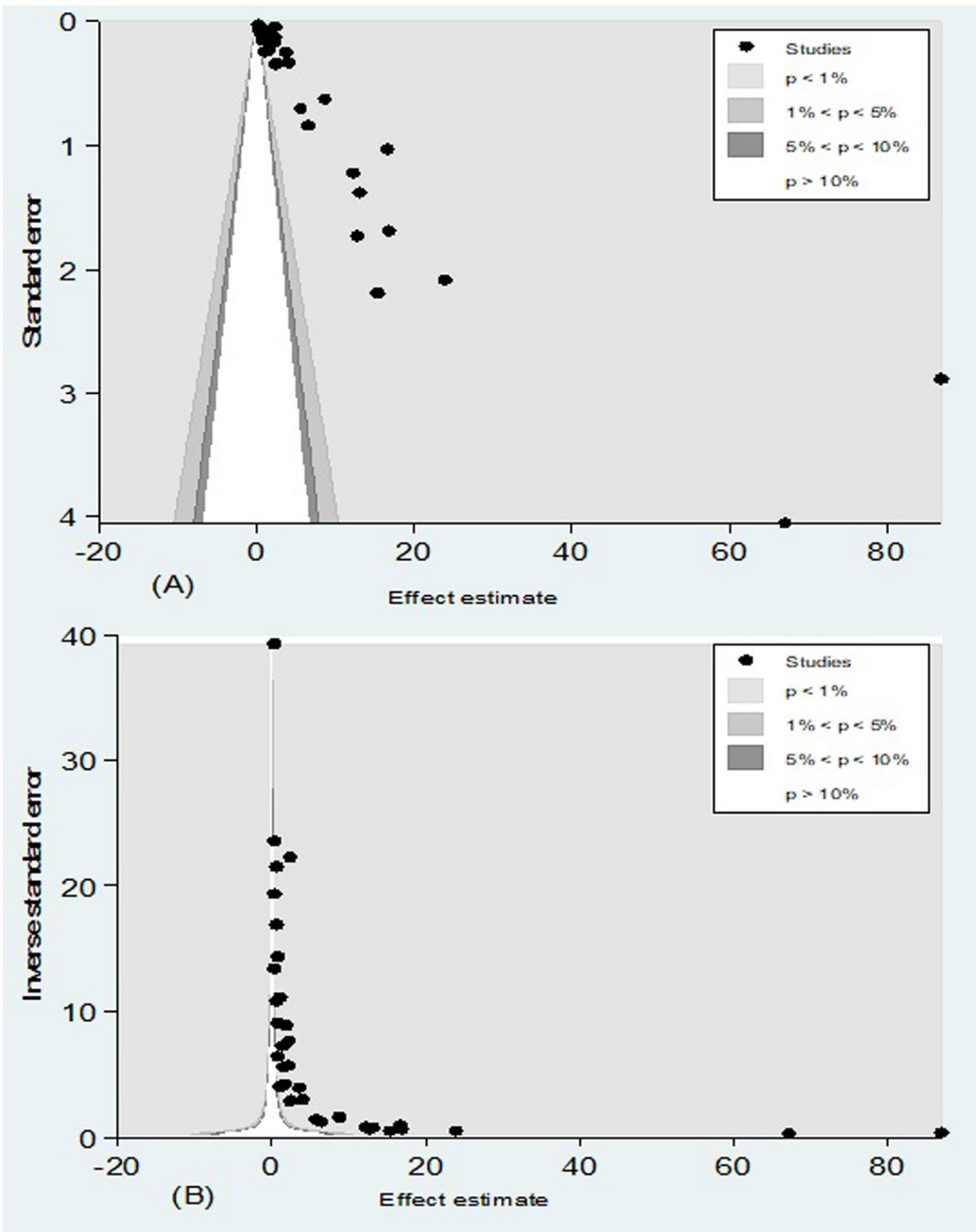

**Fig 6.** (A) Counter-enhanced, and (B) the metric inverse counter-enhanced funnel plots of publication bias for the pooled cumulative incidence of neonatal birth trauma.

delivery (vacuum extraction or forceps) was associated with 3.19 times [OR, 3.19; 95% CI, 1.92 to 5.31; Higgins and Thompson's $I^2$ statistics = 84.4%; Egger's test for small-study effects = 0.790] higher odds of neonatal birth trauma. We also found out that macrosomic neonates (defined as a birthweight of 4000grams and beyond) [67] had 5.06 times [OR, 5.06; 95% CI, 2.76 to 9.29; Higgins and Thompson's $I^2$ statistics = 43.6%; Egger's test for small-study effects = 0.061] higher odds of experiencing birth trauma (**Table 6**).

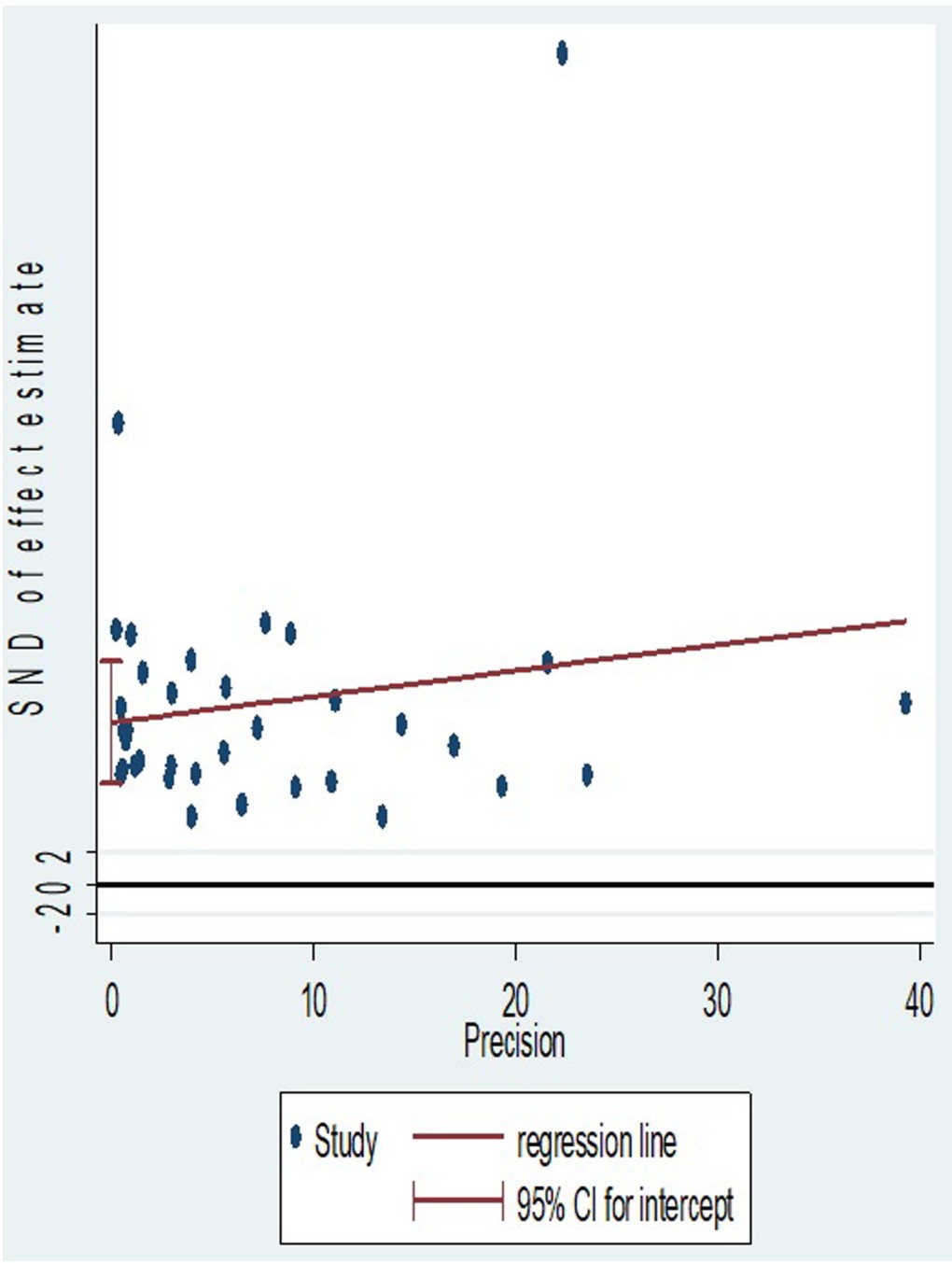

**Fig 7. Regression graph of neonatal birth trauma.**

## Discussion

Globally, neonatal health has become an important public health concern. This systematic review and meta-analysis provides the most comprehensive and granular estimation, temporal trends, clinical spectrum, and immediate adverse outcomes of neonatal birth trauma in LMICs. Our analysis found that the pooled cumulative incidence of neonatal birth trauma was estimated at 34 per 1,000 live births. The combined estimates from African countries ranked

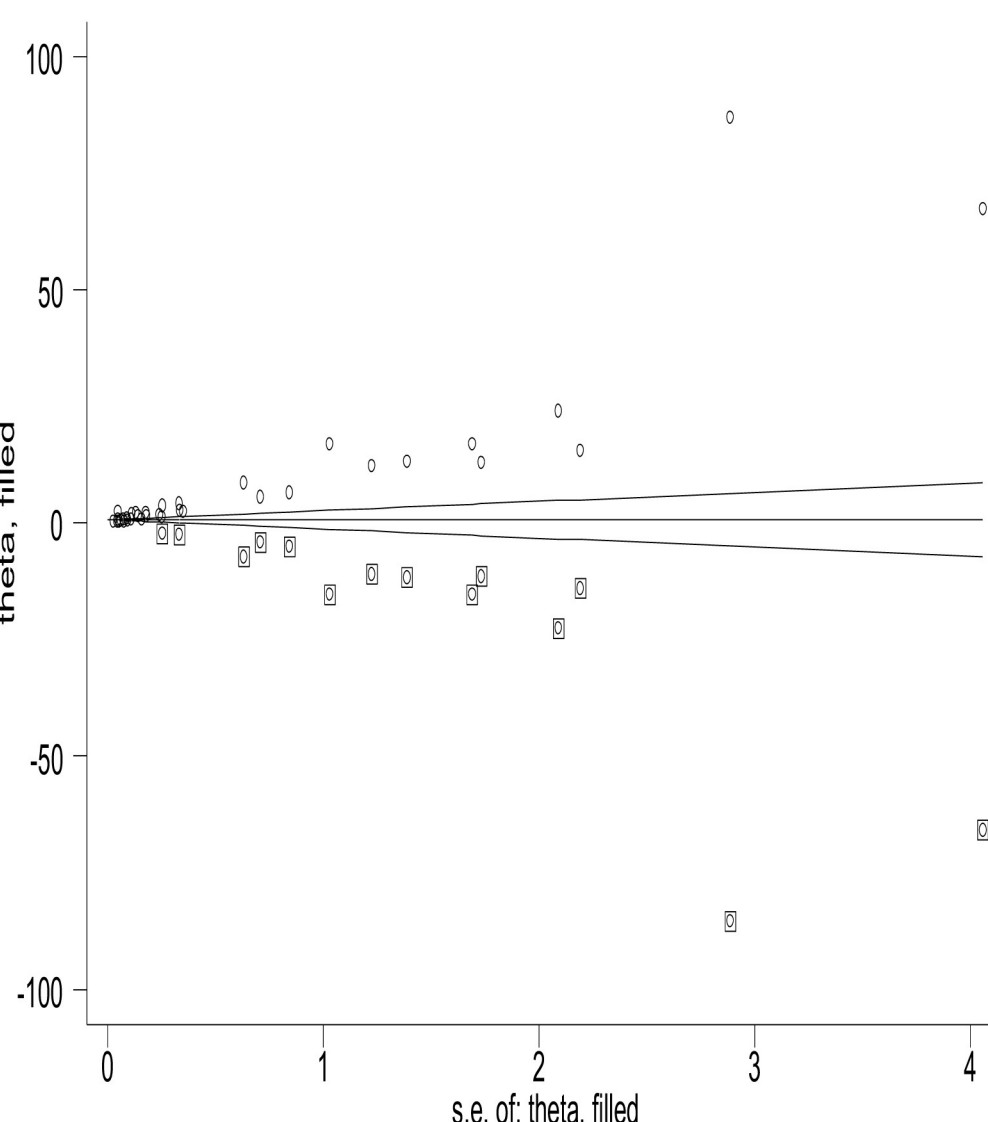

**Fig 8. Trim and fill analysis for the cumulative incidence of neonatal birth trauma.**

in first place at 52.9 per 1,000 live births. The overall incidence proportion far outweighs the incidence of birth trauma in Canada (9.6 per 1,000 live births) [17], Finland (16.6 per 1,000 live births) [13], and the US (31.1 per 1,000 live births) [15].

Factors that explain a higher burden of neonatal birth trauma in LMICs include limited hospital capabilities with modern obstetrics and neonatal care; insufficient commitments towards ensuring the implementation of international and local policies related to child health promotion; geographic inaccessibility to prenatal care; inadequate skilled health personnel; frequent application of instrumental delivery; economic depression, political instability in some regions and variation in sample size. Nevertheless, the present analysis result was lower than a Chinese study (42.9 per 1,000 live births) [16]; and Indonesia (69.2 per 1,000 live births) [34]. This may be explained in part by the fact that forceps were applied in a larger proportion

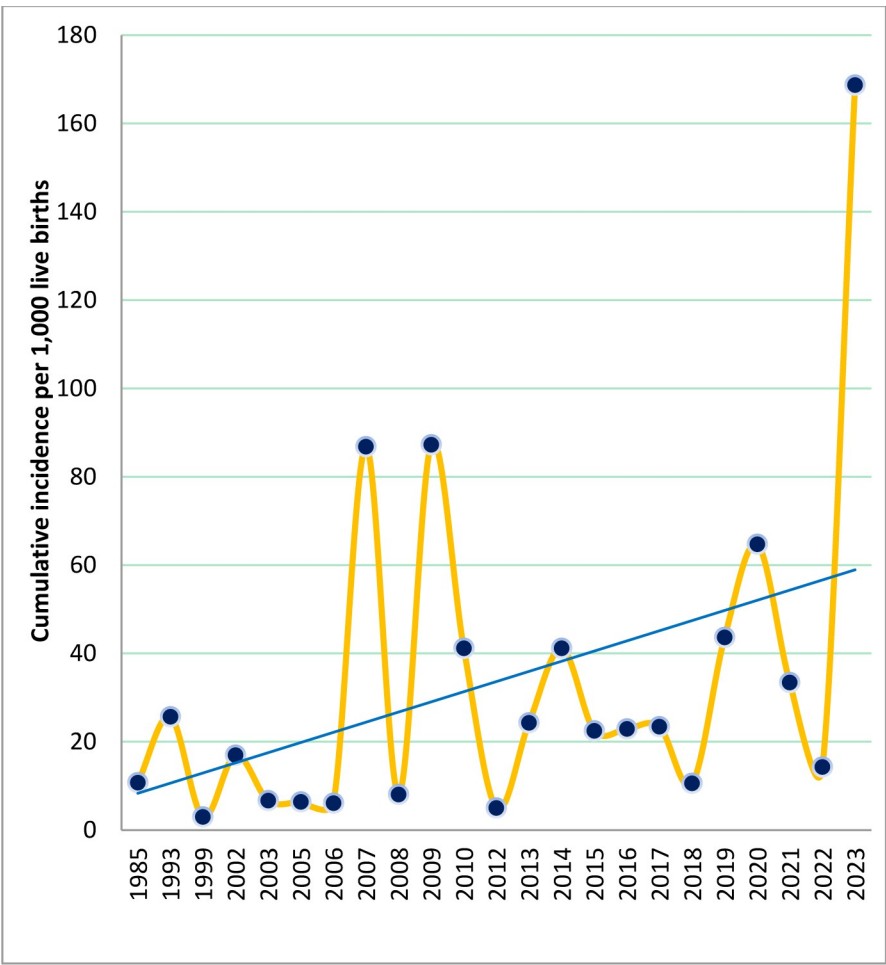

**Fig 9. The trend of neonatal birth trauma incidence in LMICs (1985 to 2023).**

(18%) in the Chinese (18%) and the Indonesian (50%) studies. Another possible justification could be due to proper registration and good record-keeping practices, and the evaluation of newborns by neonatologists right away after birth contributed to the increased and true figure of neonatal birth trauma compared to the preceding research in China.

Understanding trends of neonatal birth trauma over time is warranted to accurately inform policies and set priorities for implementation. Taking into account the existence of individual countries' variations, an overall upward trend of neonatal birth trauma was observed in the LMICs in contrast to most resource-rich countries [13–15]. This could be due to the majority (60%) of the studies were represented from Ethiopia where the estimate of the condition was highest according to country-based analyses. Moreover, inadequate access to gynecologic and prenatal care may partially explain the significant variation in the trend.

Regarding the clinical spectrum of the condition, studies on neonates from different settings show diverse birth trauma patterns [18–24, 26–28, 30, 36, 46–60, 63–66]. The most incident case of birth trauma was cephalohematoma. Such a pattern of neonatal birth trauma was generally consistent with other studies [15, 16]. According to Gupta *et al.*, cephalohematomas might be markers of morbid brain injury [15]. Furthermore, our analysis shows that the trends of neonatal birth trauma were increasing over time. In contrast, the trends have been progressively decreasing over time in resource-rich settings [13–17, 34].

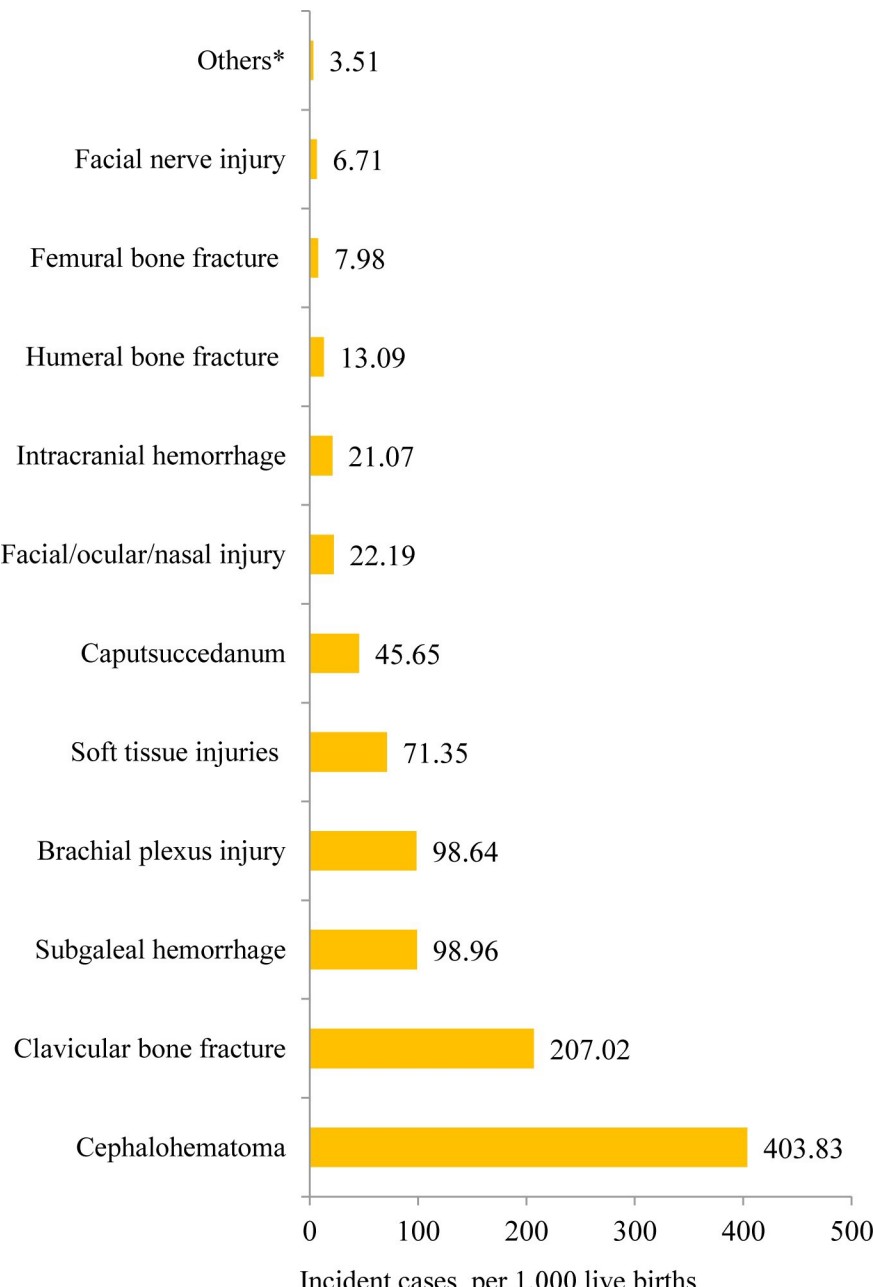

**Fig 10. The patterns of neonatal birth trauma.** * others include visceral injuries (spleen, and liver).

According to Presser et al. [68], major neonatal birth trauma is associated with increased risk of neonatal morbidity and mortality. In our study, neonatal death incidence of 5.3% [46] to 28% [47] had occurred due to birth-associated trauma, a proportion higher than reported in the US [15]. Other short-term adverse neonatal outcomes such as seizures, prolonged hospital stay, encephalopathy, anemia, and hyperbilirubinemia were also reported in studies in Canada [59], and China [16].

This systematic review and meta-analysis identified that place of residence, prolonged labor, shoulder dystocia, operative vaginal delivery, and macrosomia were associated with

**Table 5. Short-term/immediate adverse outcomes of neonatal birth trauma.**

| Authors /reference | Sample size | Event | Key outcomes |
|---|---|---|---|
| Yemane et al [26] | 717 | 88 | Ten (11%) of the neonates died from refractory hypovolemic shock ascribed to subgaleal hemorrhage. An estimated 40 (45.5%) newborns had medical problems, including anemia in ten (11.4%) and hyperbilirubinemia in 11 (12.5%). Transfusion was required for 16 (64%) neonates with anemia of acute blood loss. |
| Abdul-Mumin et al [46] | 5590 | 205 | Twenty-three (11.22%) neonates died in which extracranial birth injuries were the sole determinants. Furthermore, the hospital stay lasted up to a month. |
| Pius et al [49] | 1071 | 61 | Sepsis, hyperbilirubinemia, and anemia were adverse outcomes exhibited. |
| West and Okari [50] | 5692 | 39 | Three (7.7%) neonates died in the health facility. |
| Njokanma and Kehinde [52] | 2941 | 50 | In the hospital, death occurred in 14 (28%) of newborns. |
| Emeka et al [51] | 1735 | 19 | One (5.3%) neonate died of overwhelming sepsis secondary to infected cephalohematoma. |
| Linder et al [59] | 118,280 | 2876 | Prolonged hospitalization; neurologic features like myoclonic seizure, hypoxic-ischemic encephalopathy; and hyperbilirubinemia necessitating phototherapy. |

increased odds of neonatal birth trauma. To begin with, neonates born to mothers who were living in rural settings had 61% higher odds of sustaining birth trauma. This was congruent with a study in the Netherlands [69] that described a longer travel time from home in a rural area to a health facility associated with increased neonatal adverse outcomes. This may be due to access to maternity care is often limited in rural settings.

Prolonged labor was associated with a five-fold increased risk of sustaining birth trauma in neonates. Results supporting the current findings have been reported by Gupta *et al.* [15], and Pressler *et al.* [68]. This can be explained by the fact that there is an increased propensity to

**Table 6. Factors associated with neonatal birth trauma among neonates in low and middle-income countries.**

| Characteristics | No. of studies | Pooled odds ratio (95% confidence interval) | P.value | Statistical heterogeneity | | Egger's test |
|---|---|---|---|---|---|---|
| | | | | I² (%) | P.value | P > \| t \| |
| Mode of delivery | | | | | | |
| Operative vaginal vs. cesarean delivery [25–31, 33, 36] | 9 | 3.19 (1.92,5.31) | < 0.001* | 84.4 | <0.001 | 0.790 |
| Antenatal care follow-up | | | | | | |
| No vs. yes [29, 30, 32] | 3 | 0.62 (0.14, 2.71) | 0.523 | 93.1 | <0.001 | 0.645 |
| Neonatal birth weight | | | | | | |
| Macrosomic vs. <4000 grams [26, 29, 30, 36] | 4 | 5.06 (2.76, 9.29) | < 0.001* | 43.6 | 0.150 | 0.061 |
| Labor | | | | | | |
| Prolonged vs normal [27, 29, 31, 36] | 4 | 5.45 (2.30, 9.91) | < 0.001* | 71.8 | 0.014 | 0.498 |
| Place of residence | | | | | | |
| Rural vs. urban [25, 26, 30] | 3 | 1.61 (1.18, 2.21) | 0.003* | 0.0 | 0.723 | 0.110 |
| Fetal presentation at delivery | | | | | | |
| Non-vertex vs vertex [25, 26, 30, 31, 36] | 5 | 4.70 (1.75, 12.66) | 0.002* | 89.2 | <0.001 | 0.885 |
| Shoulder dystocia | | | | | | |
| Yes vs. no [28, 29, 31] | 3 | 6.11 (3.84, 9.74) | < 0.001* | 0.0 | 0.494 | 0.358 |

* Indicates statistically significant at P ≤ 0.05

apply forceps or vacuum when labor is prolonged to expedite the delivery of the neonate, provided that the prerequisites are fulfilled and indications exist for operative vaginal delivery.

Another important factor associated with neonatal birth trauma was shoulder dystocia. According to the literature, shoulder dystocia has been reported to complicate 0.2–3% of all vaginal deliveries and neonatal morbidities such as brain damage, brachial plexus injury, and clavicular/humeral fracture) [70, 71]. This meta-analysis revealed that the odds of sustaining birth trauma were about 6 folds higher in labor with shoulder dystocia. Other studies also buttress the current results [72, 73]. Although most cases of shoulder dystocia are unpredictable, fetal macrosomia and maternal diabetes mellitus are the most frequently cited contributing factors which in turn results in mechanical injury to the newborn [74].

Furthermore, our study identified that macrosomic neonates had five times higher odds of experiencing birth trauma. The finding was congruent with a research report in another setting [34]. Evidence suggests that macrosomic infants are at increased risk of experiencing shoulder dystocia during vaginal delivery which in turn culminates in limb fracture and brachial plexus injury [75, 76]. Besides, an elevated need for an intensive care unit admission and clavicular fracture were among the adverse events following the delivery of macrosomic neonates [77]. Spontaneous vaginal delivery of macrosomic neonates is challenging because of cephalopelvic disproportion during which time the newborn infant sustains mechanical compression or is traumatized when instruments are applied to expedite the delivery.

Lastly, this study also found that operative vaginal delivery (forceps or vacuum-assisted vaginal delivery) had three times higher odds of neonatal birth trauma compared to cesarean delivery. Our finding was congruent with a systematic review and meta-analysis report by Woldegeorgis et al. [78] that described operative vaginal delivery as a significant contributor to trauma to both the mother and newborn. Moreover, comparative studies in California and Quebec [79], the Cochrane Database of systematic review [80], a study in China [16], Bulgaria [81], and a retrospective review of operative delivery, in Singapore [82] identified that forceps or vacuum-assisted vaginal delivery was associated with increased risk of neonatal birth trauma although the risks are generally instrument specific and also affected by correct application and delivery technique as well as complex procedures. According to Mazza et al., [83] the incidence of serious neonatal birth trauma decreased to zero for consecutive 15 months in all obstetrics facilities following a significant reduction in the use of vacuum and forceps delivery shedding a green light on the importance of the establishment of the interdisciplinary team that monitors best practices.

## Strengths and limitations of the study

To the best of our knowledge, however, this is a novel study that provides comprehensive and accurate evidence of neonatal birth trauma in LMICs. Methodologically, the study was adequate and avoided duplication of similar work; intensive and comprehensive literature searches were conducted to minimize the risk of publication bias; and a double-blinded comprehensive search was conducted over a reputable period in more than six online databases to avoid missing published studies. The newly amended JBI critical appraisal tool was used for quality assessment. Further analyses were conducted to explore sources of dissemination or publication biases. Besides, a large number of neonates took part in the study, which enabled the determination of the true estimate and the investigation of the factors associated with birth trauma. This study has some limitations. To begin with, significant statistical heterogeneity was observed and therefore this requires a cautious interpretation of the result. Furthermore, only a few publications from resource-rich countries were found to compare with our results.

Lastly, the combined estimates were compared with primary studies because of the lack of a previously published meta-analysis.

## Conclusion and recommendations

The findings of our study demonstrated that the incidence of neonatal birth trauma in LMICs was considerably high. Being born to mothers living in rural areas, prolonged labor, macrosomia, operative vaginal delivery, fetal malpresentation, and shoulder dystocia were important factors contributing to neonatal birth trauma. Furthermore, there has been an increase in the temporal trends of birth trauma, related morbidities, and mortality. Anticipations and early identification of these risk factors, and prompt obstetrician's decision on the most favorable mode of delivery would help significantly decrease neonatal birth trauma and associated morbidities and mortality.

## Implications for researches and policies

Results obtained from this study have insightful implications for SDG 3.2, which aims to end preventable deaths of newborns. Besides, it helps guide health administrators and program managers at different levels, child health advocates, health care providers in health facilities, as well as the different partners and actors contributing to the implementation of policies and strategies toward the reduction of neonatal morbidities and mortalities in LMICs.

## Supporting information

**S1 Checklist. PRISMA 2020 checklist.**
(DOCX)

**S1 Table.** JBI's critical appraisal tools: (A) Descriptive cross-sectional studies. (B) Analytical cross-sectional studies. (C) case-control studies.
(DOCX)

**S1 File. Data extraction sheet.**
(XLSX)

**S2 File. Determinant factors for neonatal birth trauma.**
(XLSX)

**S3 File. Pattern and trend of neonatal birth trauma.**
(XLSX)

## Acknowledgments

We would like to express our gratitude to the authors of the original papers that were included in this systematic review and meta-analysis

## Author Contributions

**Conceptualization:** Beshada Zerfu Woldegeorgis.

**Data curation:** Beshada Zerfu Woldegeorgis, Amanuel Yosef Gebrekidan, Gizachew Ambaw Kassie, Gedion Asnake Azeze, Yordanos Sisay Asgedom, Henok Berhanu Alemu, Mohammed Suleiman Obsa.

**Formal analysis:** Beshada Zerfu Woldegeorgis, Amanuel Yosef Gebrekidan, Gizachew Ambaw Kassie, Gedion Asnake Azeze, Yordanos Sisay Asgedom, Henok Berhanu Alemu, Mohammed Suleiman Obsa.

**Investigation:** Beshada Zerfu Woldegeorgis, Amanuel Yosef Gebrekidan, Gizachew Ambaw Kassie, Gedion Asnake Azeze, Yordanos Sisay Asgedom, Henok Berhanu Alemu, Mohammed Suleiman Obsa.

**Methodology:** Beshada Zerfu Woldegeorgis, Amanuel Yosef Gebrekidan, Gizachew Ambaw Kassie, Gedion Asnake Azeze, Yordanos Sisay Asgedom, Henok Berhanu Alemu, Mohammed Suleiman Obsa.

**Project administration:** Beshada Zerfu Woldegeorgis, Amanuel Yosef Gebrekidan, Gizachew Ambaw Kassie, Gedion Asnake Azeze, Yordanos Sisay Asgedom, Henok Berhanu Alemu, Mohammed Suleiman Obsa.

**Resources:** Beshada Zerfu Woldegeorgis, Amanuel Yosef Gebrekidan, Gizachew Ambaw Kassie, Gedion Asnake Azeze, Yordanos Sisay Asgedom, Henok Berhanu Alemu, Mohammed Suleiman Obsa.

**Software:** Beshada Zerfu Woldegeorgis, Amanuel Yosef Gebrekidan, Gizachew Ambaw Kassie, Gedion Asnake Azeze, Yordanos Sisay Asgedom, Henok Berhanu Alemu, Mohammed Suleiman Obsa.

**Supervision:** Beshada Zerfu Woldegeorgis, Amanuel Yosef Gebrekidan, Gizachew Ambaw Kassie, Gedion Asnake Azeze, Yordanos Sisay Asgedom, Henok Berhanu Alemu, Mohammed Suleiman Obsa.

**Validation:** Beshada Zerfu Woldegeorgis, Amanuel Yosef Gebrekidan, Gizachew Ambaw Kassie, Gedion Asnake Azeze, Yordanos Sisay Asgedom, Henok Berhanu Alemu, Mohammed Suleiman Obsa.

**Visualization:** Beshada Zerfu Woldegeorgis, Amanuel Yosef Gebrekidan, Gizachew Ambaw Kassie, Gedion Asnake Azeze, Yordanos Sisay Asgedom, Henok Berhanu Alemu, Mohammed Suleiman Obsa.

**Writing – original draft:** Beshada Zerfu Woldegeorgis.

**Writing – review & editing:** Beshada Zerfu Woldegeorgis, Amanuel Yosef Gebrekidan, Gizachew Ambaw Kassie, Gedion Asnake Azeze, Yordanos Sisay Asgedom, Henok Berhanu Alemu, Mohammed Suleiman Obsa.

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
