## [Decision Letter · Decision Letter 0]

2 Jan 2024

PONE-D-23-28607Neonatal birth trauma and associated factors in low and middle-income countries: A systematic review and meta-analysisPLOS ONE

Dear Dr. Woldegeorgis,

Thank you for submitting your manuscript to PLOS ONE. After careful consideration, we feel that it has merit but does not fully meet PLOS ONE’s publication criteria as it currently stands. Therefore, we invite you to submit a revised version of the manuscript that addresses the points raised during the review process.

We look forward to receiving your revised manuscript.

Kind regards,

Mohammed Feyisso Shaka, MPH

Academic Editor

PLOS ONE

Journal Requirements:

Did you know that depositing data in a repository is associated with up to a 25% citation advantage (https://doi.org/10.1371/journal.pone.0230416)? If you’ve not already done so, consider depositing your raw data in a repository to ensure your work is read, appreciated and cited by the largest possible audience. You’ll also earn an Accessible Data icon on your published paper if you deposit your data in any participating repository (https://plos.org/open-science/open-data/#accessible-data).

5. We are unable to open your Supporting Information file [Supporting information.rar]. Please kindly revise as necessary and re-upload.

**Additional Editor Comments:**

Please find the reviewers reports on your manuscript below and address them accordingly.  

Reviewers' comments:

Reviewer's Responses to Questions

**Comments to the Author**

1. Is the manuscript technically sound, and do the data support the conclusions?

Reviewer #1: Yes

Reviewer #2: Yes

2. Has the statistical analysis been performed appropriately and rigorously? 

Reviewer #1: Yes

Reviewer #2: No

3. Have the authors made all data underlying the findings in their manuscript fully available?

Reviewer #1: Yes

Reviewer #2: Yes

4. Is the manuscript presented in an intelligible fashion and written in standard English?

Reviewer #1: Yes

Reviewer #2: No

5. Review Comments to the Author

Reviewer #1: The manuscript is technically sound with details methods and appropriate use of statistical analysis. The authors clearly stated their justification for the study, objectives and research questions. The overall conclusion of the study was drawn from the results.

Reviewer #2: Peer Review Report

Title: Neonatal birth trauma and associated factors in low and middle-income countries: A systematic review and meta-analysis

General Comments:

The manuscript titled "Neonatal birth trauma and associated factors in low and middle-income countries: A systematic review and meta-analysis" provides valuable insights into neonatal birth trauma in low and middle-income countries. The language and organization of the manuscript are clear and concise, making it accessible to a wide audience. However, there are a few areas where additional information or clarification would enhance the manuscript's impact. With some major revisions and additions, this manuscript has the potential to make a valuable contribution to the field.

Specific Comments:

Abstract:

a. Could you provide more information on the methods used to search for articles and select studies for inclusion in the meta-analysis? This would help readers understand the rigor of the study selection process.

b. The abstract mentions the identification of potential contributing factors for birth trauma. Could you elaborate on the specific factors identified and their respective effect sizes? Providing this information would give readers a clearer understanding of the magnitude and significance of the identified factors.

Introduction and Background:

a. Can the authors provide a more comprehensive overview of the current state of neonatal birth trauma in low and middle-income countries? It would be helpful to include statistics or prevalence rates, if available. Additionally, are there any specific regions or populations within these countries that are more affected? Providing this information would give readers a better context for understanding the significance of the study.

b. It would be beneficial to include a discussion on the impact of birth trauma on neonatal morbidity and mortality in low and middle-income countries, as well as the economic burden associated with it. This discussion would help illustrate the broader implications of the research findings.

Methods:

a. Please provide more details on the search strategy used to identify relevant articles. Were any specific inclusion or exclusion criteria applied? Including this information would enhance the transparency and reproducibility of the study.

b. How was the quality of the included studies assessed? Were any measures taken to address potential bias or heterogeneity among the studies? Describing the quality assessment process and any steps taken to address bias or heterogeneity would strengthen the study's methodology.

c. Can the authors provide a rationale for selecting the Dersimonian-Laird model as the random-effects model for the meta-analysis? Justifying the choice of this model would help readers understand the statistical approach employed.

Results:

a. The authors report a high pooled cumulative incidence of birth trauma in low and middle-income countries. Can they discuss the potential reasons for this high incidence and how it compares to rates in high-income countries? Exploring the factors contributing to the high incidence and comparing it to rates in high-income countries would provide valuable insights into the unique challenges faced by low and middle-income countries.

b. It would be interesting to know if there are any temporal trends in the incidence of birth trauma in these countries. Did the authors find any studies reporting changes over time? Including information on temporal trends would enhance the understanding of the evolving situation and potential improvements over time.

c. The factors associated with neonatal birth trauma mentioned in the results are important. However, it would be beneficial to discuss the clinical implications of these factors and how they can inform preventive strategies in low and middle-income countries. Providing this discussion would help bridge the gap between research findings and practical applications.

Discussion:

a. The discussion could be strengthened by providing a more detailed analysis of the identified risk factors and their implications. Are there any modifiable factors that could be targeted to reduce the incidence of birth trauma? Including this analysis would provide actionable insights for healthcare providers and policymakers.

b. What are the potential limitations of the included studies that may affect the generalizability of the findings? Are there any gaps in the current literature that future research should address? Discussing the limitations and gaps in the literature would help readers understand the scope and applicability of the study's findings.

6. PLOS authors have the option to publish the peer review history of their article (what does this mean?). If published, this will include your full peer review and any attached files.

Reviewer #1: No

Reviewer #2: **Yes: **Fentahun Adane Nigat, Ph.D.

---

## [Author Response · Author response to Decision Letter 0]

5 Jan 2024

General note for the academic editor regarding the overall progress

First and foremost, I am thrilled to extend my heartfelt gratitude to Mohammed Feyisso Shaka, the academic editor at PLOS ONE, for allowing us sufficient time to revise and address all of the reviewer’s concerns, comments, suggestions, and journal requirements.

Dear Mohammed Feyisso Shaka, the reviewer's comments were critically evaluated, and necessary corrections and amendments were made. The manuscript was checked to meet PLOS ONE's style requirements, and the data availability statement was updated. We thoroughly addressed specific points made and uploaded a marked-up copy (revised manuscript with tracked changes) as a "Revised Article with Changes Highlighted” file and an unmarked version of the revised paper without tracked changes and uploaded it as a separate file labeled 'Manuscript'. Lastly, for comparison, we responded to the reviewer’s comment under each query. I hope you and the reviewers receive a significantly improved manuscript. Thank you incredibly much!

A. Point-by-point response letter to reviewer #1

Dear Revevier, thank you very much for taking the time to review our manuscript. We seriously considered your comments and suggestions and incorporated them into the revised manuscript. We gave due attention to the typographical errors, results, discussion sections, conclusions, and recommendations so that the manuscript has significantly improved.

Reviewer : TITLE – Appropriate.

Authors: Thank you! 

Reviewer: Abstract- Well written with minor correction of typographical error (38 or 37 studied articles)

Authors: corrected , thank you

Introduction- Fairly written and justification for conducting the study was highlighted

Authors: Thank you! 

Methodology- This is quite detail and the chosen method will achieve the set-out objectives of the study. Statistical analysis was reflected in the results.

Authors: Thank you! 

Results- The results was well presented but the following observations were made:

1. Page 13, paragraph 2 line 2—Insert There were 365,547 live births instead of –About 365,547 live births.

Authors: We corrected it. Thank you 

2. Page 15 line 2—data were abstracted from 35 articles ( Is it 35 or 37 articles?).

Authors: The total numbers of articles included in the SRMA were 37. However, incidence data were extracted from 35 articles as the two articles (Hameed and Izzet, and Basiri et al ) were case-control. Data from these two studies were used in the effect measure metaanalysis. Thank you for your critical observation. 

3. Page 19 line1: from 1985-2023 but Title for the figure indicates from 2008-2023. This needs to be corrected.

Authors: Corrected ,thank you! 

4. Page 20 line 1, 8, and 10: why the use of the word about in the case of an absolute number? It is more appropriate to use these words in the derived figures eg in percentages and so on.

Authors: The reviewer suggestion is appropriate. We took appropriate measure.

5. Page 21 line 14 insert—assisted vaginal delivery

Authors: The reviewer suggestion is appropriate. We correct it, thank you!

Reviewer: DISCUSSION: The authors did not DISCUSS the findings adequately. In most cases, it most often just restated the findings and did literature review.

Authors: reviewer suggestion was appropriate, thank you we elaborated the result driven discussion emphasasing to justification of the pooled estimate and risk factors, .

Reviewer: CONCLUSION/RECOMMENDATION- Guarded recommendations were made, but however, there was also no mention of the focus future studies. 

Authors: The reviewer comment was appropriate. conclusion and recommendatios were revised as follows: “The findings of our study demonstrated that the incidence of neonatal birth trauma in LMICs was considerably high. Being born to mothers living in rural areas, prolonged labor, macrosomia, operative vaginal delivery, fetal malpresentation, and shoulder dystocia were important factors contributing to neonatal birth trauma. Furthermore, there has been an increase in the temporal trends of birth trauma, related morbidities, and mortality. Anticipations and early identification of these risk factors, and prompt obstetrician's decision on the most favorable mode of delivery would help significantly decrease neonatal birth trauma and associated morbidities and mortality.”Furthermore, Implications for research and policy were added to the the revised manuscript.

B. Point-by-point response letter to reviewer #2

Dear reviewer, thank you very much for taking the time to read our paper and for your helpful comments and suggestions. We made point-by-point responses for convenience as follows: 

Reviewer: Could you provide more information on the methods used to search for articles and select studies for inclusion in the meta-analysis? This would help readers understand the rigor of the study selection process.

Authors: The reviewer’s suggestion was appropriate. The word count was a factor that limited our narration. Nevertheless, we incorporated the comment into the revised manuscript by highlighting the search strategy and study selection procedure as follows: “Literature search strategies were developed using medical subject headings and text words related to the outcomes of the study. Following the risk of bias assessment, articles with appraisal scores of seven or more were deemed suitable to be included in the meta-analysis.” Thank you! 

Reviewer: The abstract mentions the identification of potential contributing factors for birth trauma. Could you elaborate on the specific factors identified and their respective effect sizes? Providing this information would give readers a clearer understanding of the magnitude and significance of the identified factors.

Authors: Dear reviewer, we have presented the specific risk factors (in summary form) with their respective ES (OR in our case) in the abstract. However, details were presented in the discussion section. We are very much thrilled to accept further suggestions in case we do not catch the point. Thank you! 

Introduction and Background

Reviewer: Can the authors provide a more comprehensive overview of the current state of neonatal birth trauma in low and middle-income countries? It would be helpful to include statistics or prevalence rates, if available. Additionally, are there any specific regions or populations within these countries that are more affected? Providing this information would give readers a better context for understanding the significance of the study.

Authors: We revised the introduction and have added individual estimates regarding the prevalence of the condition in Africa, Asia and Pacific, and Middle East regions (lines 99 to 103). Thank you!

Reviewer: It would be beneficial to include a discussion on the impact of birth trauma on neonatal morbidity and mortality in low and middle-income countries, as well as the economic burden associated with it. This discussion would help illustrate the broader implications of the research findings.

Authors: We highlited the burden of the neonatal birth trauma based on individual estimates from High income contires and low and middle income countries. “ Although reseach in to the economic impacts of birth trauma is lacking in LMICs, a study conducted in the United States (US) revealed that neonatal brachial plexus palsy hospital length of stay was 48% higher (3.6 days vs. 2.5 days) and hospitalization costs were twice ($4891 vs. $2241) as high as in neonates without the condition [8]”.

Methods

Reviewer: Please provide more details on the search strategy used to identify relevant articles. Were any specific inclusion or exclusion criteria applied? Including this information would enhance the transparency and reproducibility of the study.

Authors: The reviewer note is appropriate.The search strategy involved databases and reference list. For the database search serch terms, MeSH,Boolean operators,trucncation,and astrics were employed where necessary (Table 1) .Furthermore, we scanned reference lists of included articles to ensure literature saturation. In addition, filterations were applied. Dear reviewer, to enhance the transparency and reproducibility of the study,as you said, we re-wrote the specific paragraph ,as there were fragmentation in expression.. Thank you ! 

Reviewer: How was the quality of the included studies assessed? Were any measures taken to address potential bias or heterogeneity among the studies? Describing the quality assessment process and any steps taken to address bias or heterogeneity would strengthen the study's methodology.

Authors: Dear reviewer, regarding the Quality/risk of bias assessment, articles were subjected to appraisal using the newly amended JBI tool (supplentary file S1 Table). We considered a score of ≥7 for inclusion in the quantitative synthesis and this has been narrated in our report. dear reviewer, we are thrilled to respond in the subsequent revision in case if we haven’t got your point. 

Reviewer: Can the authors provide a rationale for selecting the Dersimonian-Laird model as the random-effects model for the meta-analysis? Justifying the choice of this model would help readers understand the statistical approach employed.

Authors: because a considerable statistical heterogeneity was observed (I2 = 99.3% and P < 0.001) in the fixed effect model, we choose the REM . In the REM ,the precesion was significantly improved although variation in the effect size attributable to heterogeneity was almost same. We incorporated the rationale in the revised manuscript. Thank you for your critical observation.

Results and discussion 

Reviewer: The authors report a high pooled cumulative incidence of birth trauma in low and middle-income countries. Can they discuss the potential reasons for this high incidence and how it compares to rates in high-income countries? Exploring the factors contributing to the high incidence and comparing it to rates in high-income countries would provide valuable insights into the unique challenges faced by low and middle-income countries.

Authors: Yes! limited hospital capabilities with modern obstetrics and neonatal care; insufficient commitments towards ensuring the implementation of international and local policies related to child health promotion; geographic inaccessibility to prenatal care; inadequate skilled health personnel; and variation in sample size, economic depression,and political instability in some regions were possible justification for such a high magnitude of neonatal birth trauma in LMICs.

Reviewer: It would be interesting to know if there are any temporal trends in the incidence of birth trauma in these countries. Did the authors find any studies reporting changes over time? Including information on temporal trends would enhance the understanding of the evolving situation and potential improvements over time.

Authors: we thoroughly read each studies conducted in LMICs regarding birth traumas. We come across individual birth type with corresponding frequencies,but not their patterns. however, the US study and other high income countries report delineated the burden of the condition over time. Therefore, we tried to present temporal trends of cumulative incidence of birth trauma ,and we hope it will be a springboard for the future studies and also help formulate implementation strategies to reverse the steadily increasing burden of neonatal birth trauma in LMICs.

Reviewer: The factors associated with neonatal birth trauma mentioned in the results are important. However, it would be beneficial to discuss the clinical implications of these factors and how they can inform preventive strategies in low and middle-income countries. Providing this discussion would help bridge the gap between research findings and practical applications.

Authors: There are predictable and preventable/modifiable risk factors associated with increased odds of neonatal birth trauma through provision of antenatal care, training of skilled health personnel, establishment and expansion of health facilities with adequate obstetric and neonatal care, proper intrapartum surveillance and early intervention, reduction of instrumental deliveries/ or promote supervised use up on indication. The current findings help guide health administrators and program managers at different levels, child health advocates, health care providers in health facilities, as well as the different partners and actors contributing to the implementation of policies and strategies toward the reduction of neonatal morbidities and mortalities in LMICs

Reviewer: The discussion could be strengthened by providing a more detailed analysis of the identified risk factors and their implications. Are there any modifiable factors that could be targeted to reduce the incidence of birth trauma? Including this analysis would provide actionable insights for healthcare providers and policymakers.

Authors: Dear reviewer, we dealt in the above comment. Furthermore, we made broad discussion with respect to these six risk factors : place of residence; prolonged labor, fetal presentation, shoulder dystocia, birth weight, and mode of delivery. Thank you!

Reviewer: What are the potential limitations of the included studies that may affect the generalizability of the findings? Are there any gaps in the current literature that future research should address? Discussing the limitations and gaps in the literature would help readers understand the scope and applicability of the study's findings.

Authors:Yes! significant statistical heterogeneity was observed and therefore this requires a cautious interpretation of the results. Furthermore, only a few publications from resource-rich countries were found to compare with our results. Lastly, the combined estimates were compared with primary studies because of the lack of a previously published meta-analysis

---

## [Decision Letter · Decision Letter 1]

26 Jan 2024

Neonatal birth trauma and associated factors in low and middle-income countries: a systematic review and meta-analysis

PONE-D-23-28607R1

Dear Dr. Woldegeorgis,

We’re pleased to inform you that your manuscript has been judged scientifically suitable for publication and will be formally accepted for publication once it meets all outstanding technical requirements.

Kind regards,

Mohammed Feyisso Shaka, MPH

Academic Editor

PLOS ONE

Additional Editor Comments (optional):

Reviewers' comments:

Reviewer's Responses to Questions

**Comments to the Author**

1. If the authors have adequately addressed your comments raised in a previous round of review and you feel that this manuscript is now acceptable for publication, you may indicate that here to bypass the “Comments to the Author” section, enter your conflict of interest statement in the “Confidential to Editor” section, and submit your "Accept" recommendation.

Reviewer #2: All comments have been addressed

2. Is the manuscript technically sound, and do the data support the conclusions?

Reviewer #2: Yes

3. Has the statistical analysis been performed appropriately and rigorously? 

Reviewer #2: Yes

4. Have the authors made all data underlying the findings in their manuscript fully available?

Reviewer #2: Yes

5. Is the manuscript presented in an intelligible fashion and written in standard English?

Reviewer #2: Yes

6. Review Comments to the Author

Reviewer #2: (No Response)

7. PLOS authors have the option to publish the peer review history of their article (what does this mean?). If published, this will include your full peer review and any attached files.

Reviewer #2: **Yes: **Fentahun Adane Nigat, PhD

---

## [Editor Report · Acceptance letter]

2 Mar 2024

PONE-D-23-28607R1 

PLOS ONE

Dear Dr. Woldegeorgis, 

I'm pleased to inform you that your manuscript has been deemed suitable for publication in PLOS ONE. Congratulations! Your manuscript is now being handed over to our production team.

Kind regards, 

on behalf of

Mr. Mohammed Feyisso Shaka 

Academic Editor

PLOS ONE